# DO NOT TRAIN IT: A LINEAR NEURAL ARCHITECTURE SEARCH OF GRAPH NEURAL NETWORKS

## ABSTRACT

Neural architecture search (NAS) for Graph neural networks (GNNs), called NAS-GNNs, has achieved significant performance over manually designed GNN architectures. However, these methods inherit issues from the conventional NAS methods, such as high computational cost and optimization difficulty. More importantly, previous NAS methods have ignored the uniqueness of GNNs, where the non-linearity has limited effect. Based on this, we are the first to theoretically prove that a GNN fixed with random weights can obtain optimal outputs under mild conditions. With the randomly-initialized weights, we can then seek the optimal architecture parameters via the sparse coding objective and derive a novel NAS-GNNs method, namely neural architecture coding (NAC). Consequently, our NAC holds a no-update scheme on GNNs and can efficiently compute in linear time. Empirical evaluations on multiple GNN benchmark datasets demonstrate that our approach leads to state-of-the-art performance, which is up to $200\times$ faster and $18.8\%$ more accurate than the strong baselines.

## 1 INTRODUCTION

Remarkable progress of graph neural networks (GNNs) has boosted research in various domains, such as traffic prediction, recommender systems, etc., as summarized in (Wu et al., 2021). The central paradigm of GNNs is to generate node embeddings through the message-passing mechanism (Hamilton, 2020), including passing, transforming, and aggregating node features across the input graph. Despite its effectiveness, designing GNNs requires laborious efforts to choose and tune neural architectures for different tasks and datasets (You et al., 2020), which limits the usability of GNNs. To automate the process, researchers have made efforts to leverage neural architecture search (NAS) (Liu et al., 2019a; Zhang et al., 2021b) for GNNs, including GraphNAS (Gao et al., 2020), Auto-GNN (Zhou et al., 2019), PDNAS (Zhao et al., 2020) and SANE (Zhao et al., 2021b). In this work, we refer the problem of NAS for GNNs as *NAS-GNNs*.

While NAS-GNNs have shown promising results, they inherit issues from general NAS methods and fail to account for the unique properties of GNN operators. It is important to understand the difficulty in general NAS training (e.g., architecture searching and weight evaluation). Based on the searching strategy, NAS methods can be categorized into three types: reinforcement learning-based methods (Zoph & Le, 2017), evolutionary algorithms-based methods (Jozefowicz et al., 2015), and differential-based methods (Liu et al., 2019a; Wu et al., 2019a) Both reinforcement learning-based and evolutionary algorithm-based methods suffer from high computational costs due to the need to re-train sampled architectures from scratch. On the contrary, the weight-sharing differential-based paradigm reuses the neural weights to reduce the search effort and produces the optimal sub-architecture directly without excessive processes, such as sampling, leading to significant computational cost reduction and becoming the new frontier of NAS.

However, the weight sharing paradigm requires the neural weights to reach optimality so as to obtain the optimal sub-architecture based on its bi-level optimization (BLO) strategy (Liu et al., 2019a), which alternately optimizes the network weights (outputs of operators) and architecture parameters (importance of operators). First, it is hard to achieve the optimal neural weights in general due to the curse of dimensionality in deep learning, leading to unstable searching results, also called the optimization gap (Xie et al., 2022). Second, this paradigm often shows a sloppy gradient estimation (Bi et al., 2020a;b; Guo et al., 2020b) due to the alternating optimization, softmax-based

estimation, and unfairness in architecture updating. This type of work suffers from slow convergence during training and is sensitive to initialization due to the wide use of early termination. If not worse, it is unclear why inheriting weights for a specific architecture is still efficient—the weight updating and sharing lack interpretability.

*Is updating the GNN weights necessary? Or, does updating weights contribute to optimal GNN architecture searching?* Existing NAS-GNN methods rely on updating the weights, and in fact, all these issues raised are due to the need to update weights to the optimum. Unlike other deep learning structures, graph neural networks behave almost linearly, so they can be simplified as linear networks while maintaining superior performance (Wu et al., 2019b). Inspired by this, we find that the untrained GNN model is nearly optimal in theory. Note that the first paper on modern GNNs, i.e., GCN (Kipf & Welling, 2017a), already spotted this striking phenomenon in the experiment, but gained little attention. To the best of our knowledge, our work is the first to unveil this and provide theoretical proof. The issues mentioned before may not be as much of a concern given no weight update is needed, making NAS-GNN much simpler.

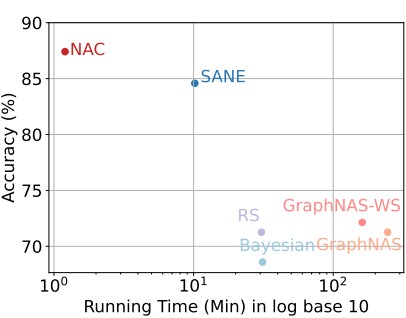

Figure 1: Accuracy vs. running time on Cora. NAC (ours) outperforms the leading methods significantly in both accuracy and speed (in minutes).

In this paper, we formulate the NAS-GNN problem as a sparse coding problem by leveraging the untrained GNNs, called neural architecture coding (NAC). We prove that untrained GNNs have built-in orthogonality, making the output dependent on the linear output layer. With no-update scheme, we only need to optimize the architecture parameters, resulting in a single-level optimization strategy as opposed to the bi-level optimization in the weight-sharing paradigm, which reduces the computational cost significantly and improves the optimization stability. Much like the sparse coding problem (Zhang et al., 2015), our goal is also to learn a set of sparse coefficients for selecting operators when treating these weights collectively as a dictionary, making sharing weights straightforward and understandable. Through extensive experiments on multiple challenging benchmarks, we demonstrate that our approach is competitive with the state-of-the-art baselines, while decreasing the computational cost significantly, as shown in Fig. 1.

In summary, our main contributions are:

- **Problem Formulation**: We present (to our best knowledge) the first linear complexity NAS algorithm for GNNs, namely NAC, which is solved by sparse coding.
- **Theoretical Analysis**: Our NAC holds a no-update scheme, which is theoretically justified by the built-in model linearity in GNNs and orthogonality in the model weights.
- **Effectiveness and Efficiency**: We compare NAC with state-of-the-art baselines and show superior performance in both accuracy and speed. Especially, NAC brings up to $18.8\%$ improvement in terms of accuracy and is $200\times$ faster than baselines.

## 2 RELATED WORK AND PRELIMINARIES

**Graph Neural Networks** (GNNs) are powerful representation learning techniques (Xu et al., 2019) with many key applications (Hamilton et al., 2017). Early GNNs are motivated from the spectral perspective, such as Spectral GNN (Bruna et al., 2014) that applies the Laplacian operators directly. ChebNet (Defferrard et al., 2016) approximates these operators using summation instead to avoid a high computational cost. GCN (Kipf & Welling, 2017b) further simplifies ChebNet by using its first order, and reaches the balance between efficiency and effectiveness, revealing the message-passing mechanism of modern GNNs. Concretely, recent GNNs aggregate node features from neighbors and stack multiple layers to capture long-range dependencies. For instance, GraphSAGE (Hamilton et al., 2017) concatenates nodes features with mean/max/LSTM pooled neighbouring information. GAT (Velickovic et al., 2018) aggregates neighbor information using learnable attention weights. GIN (Xu et al., 2019) converts the aggregation as a learnable function based on the Weisfeiler-Lehman test instead of prefixed ones as other GNNs, aiming to maximize the power of GNNs.

GNNs consist of two major components, where the *aggregation* step aggregates node features of target nodes' neighbors and the *combination* step passes previous aggregated features to networks to

generate node embeddings. Mathematically, we can update node $v$'s embedding at the $l$-th layer by

$$\boldsymbol{h}_v^l = \phi \left( \boldsymbol{W}^l \cdot o \left( \left\{ \boldsymbol{h}_v^{l-1}, \forall u \in N(v) \right\} \right) \right), \tag{1}$$

where $N(v)$ denotes the neighbours of $v$. $\boldsymbol{W}^l$ denotes the trainable weight shared by all nodes at the $l$-layer. $\phi$ is an activation function (with ReLU as default). A main difference of different GNNs lies in the design of the the aggregation functions, $o(\cdot)$. As GNNs have a large design space (You et al., 2020), this makes it challenging for the architecture search.

**Neural Architecture Search** (Xie & Yuille, 2017) generally includes three major parts, i.e., search spaces, search strategies, and evaluation methods. Recent advances adopt weight-sharing (Xie et al., 2022) to avoid (re-)training from scratch as in previous methods, where any architecture can share the weights directly for evaluation without re-training. In particular, making the weights and architectural parameters differentiable (Liu et al., 2019a; Luo et al., 2018) further improves the training efficiency. Although efficient, weight-sharing NAS has genetic issues that are caused by its bi-level optimization, including over-fitting and inaccurate gradient estimation (Xie et al., 2022), making these methods collapse unexpectedly. A variety of methods are proposed to address this from different perspectives, while mainly adding regularization towards weights, such as using $k$-shot (Su et al., 2021), paths dropout (Bender et al., 2018), sampling paths (Guo et al., 2020a), sparsity (Zhang et al., 2021a). However, these cures are still within the bi-level optimization framework, which can at best alleviate the problem but not directly tackle it. In this work, we propose NAC, a linear (i.e., single-level) optimization paradigm, to address the issues in weight-sharing NAS.

Concretely, consider WS-NAS defined by $\{\mathcal{A}, \mathcal{W}\}$, where $\mathcal{A}$ and $\mathcal{W}$ denote the search space and weights of neural networks, respectively. WS-NAS addresses two subproblems of these two space alternatively using a bi-level optimization framework (Liu et al., 2019a), including weight optimization and architecture optimization as shown in the following.

$$\boldsymbol{\alpha}^* = \underset{\boldsymbol{\alpha}}{\operatorname{argmax}} \, \mathcal{M}_{val} \left( w^*(\boldsymbol{\alpha}), \boldsymbol{\alpha} \right); \quad \text{s.t.} \quad \boldsymbol{w}^*(\boldsymbol{\alpha}) = \underset{\boldsymbol{w}}{\operatorname{argmin}} \, \mathcal{L}_{\text{train}} \left( \boldsymbol{\alpha}, \boldsymbol{w} \right), \tag{2}$$

where $\mathcal{L}$ denotes a loss function in the training set. $\mathcal{M}$ denotes an evaluation metric, such as accuracy, in the validation set. Optimizing separately, WS-NAS avoids the potential biased caused by imbalance dimensionality between $w$ and $\boldsymbol{\alpha}$, i.e., the neural weights and the architecture parameters. DARTS (Liu et al., 2019a) further converts the discrete search space into a differentiable one with the help of the softmax function that is writing as $\frac{\exp\left(\alpha_o^{(i,j)}\right)}{\sum_{o' \in \mathcal{O}} \exp\left(\alpha_{o'}^{(i,j)}\right)}$, where $\mathcal{O}$ represents the search space consisting of all operators. Thus, we can get the importance of each operator in the format of probability. However, it is difficult to reach the optimal state for the objectives defined in Eq. (2) simultaneously, because achieving optimality in deep neural networks is still an open question, and the gradient w.r.t. $\boldsymbol{\alpha}$ is intractable (Xie et al., 2022). In practice, researchers often apply early stopping to tackle it, which leads to unexpected model collapse (Zela et al., 2020) and initialization sensitivity (Xie et al., 2022). Another line of work (Chen et al., 2021) (Shu et al., 2021) uses neural tangent kernels (NTK) to search the network structure, called training-free NAS and focuses on CNN architectures. These hold strong assumptions when analyzing due to the need for infinite width of networks, and thus far from reality. In contrast, we do not have such an assumption by taking advantage of the built-in linearity in GNNs to get untrained GNNs to work.

**NAS for Graph Neural Networks:** Many NAS methods have been proposed and adapted for GNNs, such as GraphNAS (Gao et al., 2020), Auto-GNN (Zhou et al., 2019), and SANE (Zhao et al., 2021b), and thus enable learning an adaptive GNN architecture with less human effort. Methods like Auto-GNN and GraphNAS adopt reinforcement learning (RL) for searching, and thus suffer from the expensive computational cost. Thanks to the weight-sharing policy, SANE (Zhao et al., 2021b) avoids heavy retraining processes in RL methods, which leads to superior performance in general. However, this type of work also faces issues from the weight-sharing differential framework, such as bias optimization and instability due to bi-level optimization. In this work, our proposed method NAC focuses on the NAS for GNNs, and we use GraphNAS and SANE as our main baselines.

**Sparse Coding** (Zhang et al., 2015) is a well-studied topic, and its goal is to learn an informative representation as a linear combination from a collection of basis or atom, which is called a *dictionary* collectively. The standard SC formulation is written as follows:

$$\min_{\mathcal{D}, \boldsymbol{\Gamma}} \, \mathcal{L}(\mathcal{V}, \mathcal{D}, \boldsymbol{\Gamma}) = \|\mathcal{V} - \mathcal{D}\boldsymbol{\Gamma}\|_F^2, \quad \text{s.t.} \quad \forall i, \|\boldsymbol{\Gamma}_{\cdot i}\|_0 \leq \tau_0; \forall j, \|\mathcal{D}_{\cdot j}\|_2 = 1, \tag{3}$$

where $\boldsymbol{\Gamma}$ is the sparse coefficient vector, $\mathcal{D}$ is the dictionary, $\tau_0$ is a value to control the sparsity. The first constraint is to encourage a sparse representation on $\boldsymbol{\Gamma}$ and the second constraint is to normalize the atoms in the dictionary. NAS aims to find a small portion of operators from a large set of operators, making it a sparse coding problem in nature. We, therefore, reformulate the NAS-GNN problem as a sparse coding problem.

## 3 NETWORK ARCHITECTURE CODING (NAC) FOR NAS-GNN

To find the best-performing architecture, we introduce a novel NAS-GNN paradigm, Neural Architecture Coding (NAC), based on sparse Coding, which requires no update on the neural weights during training. Technical motivation for NAC stems from the observation that an untrained GNN performs well, and training it might not provide extra benefit. In the remainder of this section, we provide a theoretical analysis of why no-update GNNs are preferred (Section. 3.1) and propose a new NAS-GNN paradigm based on our theorems by leveraging untrained GNNs (Section. 3.2).

### 3.1 ANALYSIS OF NO-UPDATE SCHEME IN GNNS

In the performance analysis of GNNs, the nonlinear activation function $\phi$ defined in Eq. (1) is often skipped. Following (Wu et al., 2019b), we simplify a GNN as $\boldsymbol{F} = \boldsymbol{A} \ldots \boldsymbol{A}(\boldsymbol{A}\boldsymbol{X}\boldsymbol{W}_1)\boldsymbol{W}_2 \ldots \boldsymbol{W}_L$ $= \boldsymbol{A}^L \boldsymbol{X} \prod_{l=1}^{L} \boldsymbol{W}_l$, where $\boldsymbol{A}$ is the adjacency matrix, $\boldsymbol{W}_l$ is the neural weight at the $l$-th layer from a total number of layers, $L$. We then obtain the final output as $Out = \boldsymbol{F}\boldsymbol{W}_o$ by passing a linear layer $\boldsymbol{W}_o$. At the $t$-th iteration, we denote the GNN weight at the $l$-th layer as $\boldsymbol{W}_l(t)$ and the theoretical output layer weight as $\tilde{\boldsymbol{W}}_o(t)$. By initializing the network for all layers, i.e., $[\boldsymbol{W}_{1:L}, \boldsymbol{W}_o]$, we aim to train the network to get the optimal weights, denoted as $[\boldsymbol{W}_{1:L}^*, \boldsymbol{W}_o^*]$, where we assume the optimal weight can be obtained at $+\infty$-th iteration: $\boldsymbol{W}_l^* = \boldsymbol{W}_l(+\infty)$. We use gradient descent as the default optimizer in GNNs.

We now provide our first main theorem to investigate why an untrained GNN model can attain the same performance as the optimal one.

**Theorem 3.1.** Assume $\boldsymbol{W}_l(0)$ is randomly initialized for all $l \in [1, L]$, if $\prod_{l=1}^{L} \boldsymbol{W}_l(0)$ is full rank, there must exist a weight matrix for the output layer, i.e., $\tilde{\boldsymbol{W}}_o$, that makes the final output the same as the one from a well-trained network:

$$\boldsymbol{A}^L \boldsymbol{X} \prod_{l=1}^{L} \boldsymbol{W}_l^* \boldsymbol{W}_o^* = \boldsymbol{A}^L \boldsymbol{X} \prod_{l=1}^{L} \boldsymbol{W}_l(0)\tilde{\boldsymbol{W}}_o. \tag{4}$$

*Proof.* Because $\prod_{l=1}^{L} \boldsymbol{W}_l(0)$ is a full-rank matrix, $\prod_{l=1}^{L} \boldsymbol{W}_l(0)$ is invertible. We can define $\tilde{\boldsymbol{W}}_o$ by

$$\tilde{\boldsymbol{W}}_o \overset{\text{def}}{=} (\prod_{l=1}^{L} \boldsymbol{W}_l(0))^{-1} \prod_{l=1}^{L} \boldsymbol{W}_l^* \boldsymbol{W}_o^*, \tag{5}$$

Hence, by multiplying $\prod_{l=1}^{L} \boldsymbol{W}_l(0)$ and $\boldsymbol{A}^L \boldsymbol{X}$ on both sides in Eq. (5), we can attain Eq. (4). ☐

This theorem implies that the output layer alone can ensure the optimality of the network even when all previous layers have no updates. Next, we show that training in the standard setting of GNNs is guaranteed to reach the desired weight of the output layer, i.e. $\tilde{\boldsymbol{W}}_o$, under mild conditions. We begin by providing the theorem from (Gunasekar et al., 2018).

**Theorem 3.2** (From (Gunasekar et al., 2018))**.** Assume we have the following main conditions: 1) the loss function is either the exponential loss or the log-loss; 2) gradient descent is employed to seek the optimal solution; and 3) data is linearly separable. When defining a linear neural network as $\boldsymbol{Y} = \boldsymbol{X}\boldsymbol{W}_1\boldsymbol{W}_2.....\boldsymbol{W}_L = \boldsymbol{X} \prod_{l=1}^{L} \boldsymbol{W}_l = \boldsymbol{X}\boldsymbol{\beta}$, we always have the same optimal weight regardless of the number of layers when using gradient descent,

$$\boldsymbol{\beta}^* = \underset{\boldsymbol{\beta}}{\operatorname{argmin}} \|\boldsymbol{\beta}\|^2, \quad \text{s.t.} \quad \boldsymbol{X}\boldsymbol{\beta} \odot \boldsymbol{s} \geq \boldsymbol{1}, \tag{6}$$

where $\boldsymbol{s}$ is the ground-truth label vectors with elements as $1$ or $-1$; $\boldsymbol{1}$ denotes a vector of all ones; $\odot$ denotes the element-wise product.

This theorem suggests that the optimal weights obtained by the gradient descent are the same regardless of how many layers are in the network, which is equivalent to finding a max-margin

separating hyperplane. Following this, we prove that $\tilde{W}_o$ can be obtained based on the above conditions:

**Theorem 3.3.** Assume a GNN model has either the exponential loss or the log-loss, the desired weight $\tilde{W}_o$ is secured when updating with gradient descent and initializing $\prod_{l=1}^{L} W_l(0)$ as an orthogonal matrix. Mathematically, we have $\hat{W}_o(+\infty) = \hat{W}_o^* = \tilde{W}_o$.

*Proof.* We first frame the simplified GNNs as $A^L X \prod_{l=1}^{L} W_l = (A^L X)(\prod_{l=1}^{L} W_l)$. According to Theorem. 3.2, we define the corresponding $\beta$ as $\beta = \prod_{l=1}^{L} W_l$. When using the gradient descent to optimize $W_l$ and $W_o$, we have the optimized $\beta$ as $\beta^* = \prod_{l=1}^{L} W_l^* \cdot W_o^*$. The obtained $\beta^*$ can be viewed as the max-margin hyperplane to separate data $A^L X$ according to Theorem. 3.2. Omitting the output layer, we let the untrained GNN model as $A^L X \prod_{l=1}^{L} W_l(0) = A^L X O$ and assume $O = \prod_{l=1}^{L} W_l(0)$ is an orthogonal square matrix. Denoting a complete GNN model as $A^L X O W_o$, the optimized output weight, i.e. $\hat{W}_o^*$, becomes the max-margin separating hyperplane of $A^L X O$ because it is also trained by gradient descent. Here, we fix the $W_l(0)$ during training to maintain its orthogonality. We know that the max-margin hyperplane of any data remains the same if and only if it takes orthogonal transformations, resulting in the following equivalence: $O \hat{W}_o^* = \beta^* = \prod_{l=1}^{L} W_l^* \cdot W_o^*$. Finally, we get $\hat{W}_o^* = O^{-1} \prod_{l=1}^{L} W_l^* W_o^* = (\prod_{l=1}^{L} W_l(0))^{-1} \prod_{l=1}^{L} W_l^* W_o^* = \tilde{W}_o$. □

In summary, our proposed theorems prove that a GNN with randomly initialized weights can make the final output as good as a well-trained network, where one needs to update the overall network with gradient descent and initialize networks with orthogonal weights. Apart from the optimality, another immediate benefit of this no-update scheme is that the computational cost will be significantly reduced. This is of particular importance when the budget is limited, as the main computational cost of NAS comes from this update.

In this work, we treat the entire NAS supernet as a general GNN model where each layer is a mixture of multiple operators. The above theorems show that one can approximate the optimal output by using the network without training, where we only need to update the final linear layer of the supernet. This finding motivates us to learn architectures by taking advantage of randomly initialized networks.

We want to emphasize that real-world scenarios can break the optimal conditions, such as the high complexity of data and the early stopping of the training. Even though, our strategy still provides near-optimal performance, if not the best, in experiments.

## 3.2 Architecture Searching via Sparse Coding

Building on the above theorems, we present a novel NAS-GNN paradigm in which an untrained GNN model can also yield optimal outputs under mild conditions in the following. This optimality allows us to waive the effort to design complicated models for updating weights and makes us focus on architecture updates. At a high level, the ultimate goal of NAS is to assign the coefficients of unimportant operators as zeros and the important ones as non-zeros. Towards this, searching the optimal architecture is equivalent to finding the optimal sparse combination of GNN operators, i.e., $\alpha$. Notably, one of the most successful algorithms for this purpose is sparse coding (Olshausen & Field, 1997), which has strong performance guarantees.

*What matters most when we reformulate the NAS problem as a sparse coding problem over a dictionary defined on GNN operators?* According to the research in sparse coding (Aharon et al., 2006; Tropp, 2004; Candes et al., 2006), its performance depends on the quality of the dictionary. In particular, the most desirable property is mutual coherence or orthogonality. High incoherence in the dictionary tends to avoid ambiguity in the solution and improve optimization stability, resulting in optimal or at least sub-optimal solutions. We prove a dictionary that is defined on deep neural networks has built-in orthogonality, as shown in the theorem below.

**Theorem 3.4.** Let the neural weights of each operator in a deep neural network be an atom and stack them column-wisely, we can guarantee the existence of an orthogonal dictionary.

Please see Appendix A.2 for more details. This theorem implies that the obtained weights are qualified to be a dictionary due to the built-in high dimensionality in deep neural networks. Taking

inspiration from this, we propose a new NAS paradigm called *neural architecture coding* as follows:

$$\min_{\boldsymbol{\alpha}} \quad \mathcal{L}_{CE}\left(\boldsymbol{y}, f(\mathbf{z})\right) + \|\mathbf{z} - \mathbf{h}^L\|_F^2 + \rho\|\boldsymbol{\alpha}\|_1,$$

$$\text{where} \begin{cases} \mathbf{h}_v^l = \phi\left[\boldsymbol{W}^l \cdot \bar{o}^l\left(\left\{\mathbf{h}_u^{l-1}, \forall u \in N(v)\right\}\right)\right] \\ \bar{o}^l(x) = \boldsymbol{o}^l \hat{\boldsymbol{\alpha}}_l = \sum_{k=1}^{K} \frac{\alpha_{lk}}{||\boldsymbol{\alpha}_l||_2} o^{lk}(x) \\ \boldsymbol{o}^l = [o^{l1}(x), o^{l2}(x), \dots, o^{lK}(x)] \end{cases} \tag{7}$$

where $\rho$ is the sparsity hyperparameter, $f(\cdot)$ is a linear function as the final output layer, e.g., MLP, $\mathbf{h}$ represents all nodes' embeddings, and $\mathcal{L}_{CE}$ defines the Cross-Entropy loss function. Here, we set the number of GNN operators to $K$ and the number of hidden layers to $L$. $o^{lk}$ denotes the $k$-th operator at the $l$-th layer and its output is a vector; $\alpha_{lk}$ denotes the scalar coefficient for operator $k$ at $l$th-layer and thus $\boldsymbol{\alpha}_l \in \mathbb{R}^k$ ; $\boldsymbol{o}^l$ is the weight vector from all operators at the $l$-th layer. $\mathbf{h}_v^l$ denotes the embedding of the node $v$ at the $l$-th layer , where its neighborhood aggregation process is performed as the weighted summation of the outputs from all operators as $\bar{o}^l(x)$. $\mathbf{y}$ is the node label.

---

**Algorithm 1** The NAC algorithm

---

**Require:** The search space $\mathcal{A}$;
**Ensure:** The architecture $\boldsymbol{\alpha}$
 1: Randomly initializing $\boldsymbol{W}^l$, for $l = 1, \dots, L$; set $\boldsymbol{\alpha} = \mathbf{1}$;
 2: **while** $t = 1, \dots, T$ **do**
 3:     Performing the feature aggregation at each layer as $\bar{o}^l(x) = \boldsymbol{o}^l \hat{\boldsymbol{\alpha}}_l = \sum_{k=1}^{K} \frac{\alpha_{lk}}{||\boldsymbol{\alpha}_l||_2} o^{lk}(x)$;
 4:     Computing $\mathbf{h}_v^l = \phi\left[\boldsymbol{W}^l \cdot \bar{o}^l\left(\left\{\mathbf{h}_u^{l-1}, \forall u \in N(v)\right\}\right)\right]$;
 5:     Optimizing $\boldsymbol{\alpha}$ based on the objective function in Eq. (7) w.r.t. $\boldsymbol{\alpha}$ under the fixed dictionary $\boldsymbol{o}$ ;
 6:     Updating $\boldsymbol{W}_o$ based on the objective function in Eq. (7) w.r.t. $\boldsymbol{W}_o$ under fixed $\boldsymbol{\alpha}$;
 7: **end while**
 8: Obtain the final architecture $\{\boldsymbol{\alpha}^*\}$ from the trained $\boldsymbol{\alpha}$ via an argmax operation at each layer;

---

It is important to note that, rather than computing the optimal neural weight for $o^{lk}$ like previous NAS methods explicitly, we use randomly initialized weights instead and focus on optimizing the combination coefficient $\boldsymbol{\alpha}$, i.e. architecture parameters. The desired optimal embedding $\mathbf{z}$ is then used for downstream applications, such as node classification. In practice, we often let $\mathbf{z} = \mathbf{h}^L$ for computational convenience, which makes the loss of the second term to be zero.

The sparsity regularization $||\boldsymbol{\alpha}||_1$ allows us to rank the importance of the operator directly, alleviating inaccurate gradient estimation such as softmax (Garg et al., 2021). We want to emphasize that this regularization does not break the requirement of using gradient descent. Since the architecture, i.e., $\boldsymbol{\alpha}$, is fixed when updating the output layer weights, we can use gradient descent for updating as required, so the theorems in Section. 3.1 hold.

### 3.3 IMPLEMENTATION DETAILS AND COMPLEXITY ANALYSIS

The overall algorithm is presented at Algorithm 1. Computing NAC consists of two major parts: the forward pass and the backward pass. Given the search space, the computation of the forward is then fixed and regarded as a constant. Therefore, the computational complexity mainly focuses on the backward pass in the NAC algorithm. In summary, the algorithmic complexity of NAC is $O(T * |\boldsymbol{\alpha}|)$, where $|\boldsymbol{\alpha}|$ is the size of $\boldsymbol{\alpha}$. Please refer to Appendix A.3 for more details about the implementation.

## 4 EXPERIMENTS

We conduct experiments to address the following issues: **(1)** *How does NAC perform in comparison to the leading baselines?* **(2)** *How does the no-update scheme affect other methods?* **(3)** *Is NAC robust?* Sec. 4.2-Sec. 4.5 answer the above questions accordingly.

### 4.1 EXPERIMENT SETUP

**Datasets.** We performed experiments on transductive learning for node classification (Zhao et al., 2021b). For this setting, we use four benchmark datasets: CiteSeer (Giles et al., 1998), Cora (Kipf & Welling, 2016), PubMed (Sen et al., 2008), and Computers (McAuley et al., 2015). Also, we follow the data partition setting (training/validation/testing) as in (Zhao et al., 2021b). For more details

about transductive learning task and introduction of each dataset, please refer to Appendix A.1 for more details.

**Methods.** Following (Zhao et al., 2021b), we compare our NAC with the following strong baselines: 1) Random Search (RS) (Bergstra & Bengio, 2012) and 2) Bayesian Optimization (BO) (Jones et al., 1998), 3) GraphNAS (Gao et al., 2020): a popular reinforcement learning-based (RL) method, 4) GraphNAS-WS (Zhao et al., 2021b): a variant of GraphNAS with weight sharing, and 5) SANE (Zhao et al., 2021b). Among all baselines, the first two are hyperparameter optimization (HPO) methods (Yu & Zhu, 2020). GraphNAS and GraphNAS-WS are two popular methods following the weight sharing paradigm while SANE is the most recent work on automated GNNs using the weight-sharing differential paradigm, which is the closest one to our work. We also evaluate NAC-updating, i.e., NAC with weight updates, to compare it with the proposed one with no-update scheme.

**Search space.** We select a list of node aggregators (i.e., operators) in GNNs to form the search space. More specifically, they include the following seven aggregators: multi-layer perceptrons (MLP), GCN (Kipf & Welling, 2017b), GAT (Velickovic et al., 2018), GIN (Xu et al., 2019), Geniepath (Liu et al., 2019b), Graphsage (Hamilton et al., 2017), and ChebNet (Defferrard et al., 2016). Note that MLP is a special aggregator with fixed frequency when viewing GNNs from a spectrum perspective (Balcilar et al., 2020). Besides, Graphsage and GAT contain different aggregation functions, i.e., maximum and LSTM. We apply all these variations as operators in the experiments. Note that we do not include the layer aggregators like skip-connection in our search space. The underlying rationale is that recent research (Zela et al., 2020; Chen & Hsieh, 2020; Liang et al., 2019) finds that NAS models converge to skip-connection much faster than other operators due to its parameter-free nature. This introduces biased searching and yields poor generalization. Our experiments ensure that the search space is the same for all baselines.

We do not include layer-wise aggregators like JKNets (Xu et al., 2018) in the main experiment. As we show in Sec. 3.1, our theory is built upon the classical GNN framework, which equals matrix multiplication. JKNets cannot be represented in this framework, which contradicts our assumptions.

**Evaluation Metrics.** The proposed approaches aim to boost the ability to learn efficacious architecture for downstream tasks. The classification accuracy and the runtime (i.e., wall-clock time) are two widely used metrics in NAS research to measure the model performance and efficiency, respectively (Zhang et al., 2021a). The reported results are the average of the best results in 4 runs from different random seeds.

We set the runtime of NAC, SANE, and GraphNAS as the time they take to obtain the final architectures with 100 epochs[1]. The runtime of RS and Bayesian is fixed to the time of 100 times trial-and-error processes.

Table 1: Experimental results on the compared methods: our NAC attains superior performance in both accuracy (%) and efficiency (in minutes).

| | CiteSeer | | Cora | | PubMed | | Computers | |
| | Accuracy | Time | Accuracy | Time | Accuracy | Time | Accuracy | Time |
|---|---|---|---|---|---|---|---|---|
| RS | $70.12_{\pm2.36}$ | 14.4 | $71.26_{\pm4.68}$ | 30.6 | $86.75_{\pm0.82}$ | 187.8 | $77.84_{\pm1.35}$ | 8.75 |
| BO | $70.95_{\pm1.62}$ | 18 | $68.59_{\pm6.66}$ | 31.2 | $87.42_{\pm0.68}$ | 189.6 | $77.46_{\pm2.02}$ | 17.65 |
| GraphNAS | $68.69_{\pm1.30}$ | 253.8 | $71.26_{\pm4.90}$ | 245.4 | $86.07_{\pm0.51}$ | 1363.8 | $73.97_{\pm1.79}$ | 86.37 |
| GraphNAS-WS | $65.35_{\pm5.13}$ | 80.4 | $72.14_{\pm2.59}$ | 161.4 | $85.71_{\pm1.05}$ | 965.4 | $72.99_{\pm3.44}$ | 42.47 |
| SANE | $71.84_{\pm1.33}$ | 4.2 | $84.58_{\pm0.53}$ | 10.2 | $87.55_{\pm0.78}$ | 107.4 | $90.70_{\pm0.89}$ | 0.72 |
| NAC | $\mathbf{74.62}_{\pm0.38}$ | **1.2** | $\mathbf{87.41}_{\pm0.92}$ | **1.2** | $88.04_{\pm1.06}$ | **9.0** | $\mathbf{91.64}_{\pm0.14}$ | **0.23** |
| NAC-updating | $74.17_{\pm1.18}$ | 4.2 | $86.62_{\pm1.14}$ | 3.6 | $\mathbf{88.10}_{\pm0.86}$ | 25.8 | $90.89_{\pm1.10}$ | 0.70 |

## 4.2 COMPARISON RESULTS

Results in Table 1 and Fig. 1 show that NAC attains superior performance than all baselines regarding both *accuracy* and *efficiency*. More specifically, we observe the following.

- In terms of model performance, our NAC beats all baselines and attains up to $2.83\%$ improvement over the best baseline, i.e., SANE, while attaining up to $18.8\%$ improvement over the Bayesian method, the best HPO method. Thanks to our non-updating scheme, it prompts the outputs near the optimal to make the selection of the best-performing architecture reliable, as opposed to biased

---

[1]The running time for the Computers dataset is measured on a single GPU, which is different from all other three datasets.

optimization in others. We notice that in one case, NAC is slightly worse than NAC-updating, about $0.06\%$. This case reminds us that the optimal condition can get compromised due to the high complexity of the data. Still, NAC holds for near-best performance, demonstrating its robustness. Empirical results corroborate our theories in Section. 3.1 that non-updating scheme is preferred. The underlying assumption of RL-based and BO-based methods is to obtain an accurate estimation of the distribution of the search space. However, these methods rely on random sampling to perform the estimation, thus have no guarantee of the search quality due to the high complexity in the search space. For instance, RL-based methods are sometimes even worse than RS-based methods, pure randomly sampled, implying the unreliability of such estimation under a limited budget. WS-based methods couple the architecture search and the weight training, so they struggle to train the network to be optimal and obtain a biased network due to oscillations between the two components.

- In terms of model efficiency, our NAC achieves superior performance, around $10\times$ faster than SANE and up to $200\times$ time faster than GraphNAS. Our non-updating scheme requires nearly no weight update, thus giving NAC an incomparable advantage in reducing the computation cost. All previous NAS methods need to update neural weights, for example, RL-based costs the most due to retraining the network from scratch at each time step, and WS-based costs the least by reusing neural weights.

### 4.3 No-update Scheme at Work

To further validate our no-update scheme, we evaluate its effect on other weight-sharing methods. Since SANE attains the best performance among all baselines, we test the no-update scheme on SANE as SANE*, i.e., SANE with fixed neural weights. Results in Table 3 show that SANE* outperforms the one with updates. This result implies that we can improve the performance of NAS-GNN methods by simply fixing the weights with random initialization. This yields a much lower computational cost in the training.

Table 2: Performance comparison between SANE, NAC$^+$ and NAC.

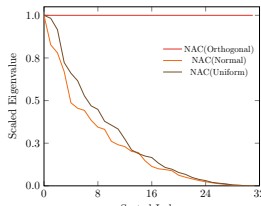

|        | CiteSeer Acc(%) | Cora Acc(%) | PubMed Acc(%) | Computers Acc(%) |
|--------|-----------------|-------------|---------------|------------------|
| SANE   | $71.84_{\pm 1.33}$ | $84.58_{\pm 0.53}$ | $87.55_{\pm 0.78}$ | $90.70_{\pm 0.89}$ |
| NAC$^+$ | $71.76_{\pm 2.08}$ | $87.27_{\pm 1.20}$ | $87.59_{\pm 0.22}$ | $91.07_{\pm 0.71}$ |
| NAC    | $\mathbf{74.62}_{\pm 0.38}$ | $\mathbf{87.41}_{\pm 0.92}$ | $\mathbf{88.04}_{\pm 1.06}$ | $\mathbf{91.64}_{\pm 0.14}$ |

Figure 2: Visualization of the sorted eigenvalues of weights of NAC with different initialization.

### 4.4 Ablation Studies

**Initialization:** Recall that in Sec. 3.1, a prerequisite is the orthogonality on the neural weights matrix. This makes the initialization critical to the final performance. To investigate the effect of initialization, we try three types of initialization methods, including normal, uniform, and orthogonal. Table 4 shows that NAC with orthogonal initialization outperforms other two cases of initialization. The results again confirm our theory.

To get a better understanding of the reason, we visualize the eigenvalues of $\prod_{l=1}^{L} \boldsymbol{W}_l$ on different initialization cases in Figure 2. The results show that the eigenvalues of NACs with normal and uniform initialization decay accordingly. This makes the input feature tends to project to the subspace spanned by the top-ranked eigenvectors, which results in feature distortion and leads to performance degeneration. In contrast, when NAC with orthogonal initialization, the eigenvalues are distributed uniformly, which ensures the spanned space does not bias toward any particular eigenvectors.

In summary, NAC with orthogonal initialization attains the best performance and confirms our theory. In addition, results from non-orthogonal initializations are still better than other baselines, demonstrating the robustness of NAC.

**The linear output layer:** Recall that for a given linear output layer, there exists the optimal weight, i.e. $\tilde{W}_o$, to secure the optimal output according to Theorem. 3.1. Nevertheless, obtaining optimal weights in a deep learning algorithm is an open question due to its high dimensionality. Often, in practice, we train the network with early stopping, especially in NAS methods. Therefore, obtaining the optimal weights for the output layer is not possible. *Is training the output layer necessary?* This

Table 3: Comparison between SANE and SANE* (w/o. weight updates).

| | CiteSeer Acc(%) | Cora Acc(%) | Pubmed Acc(%) | Computers Acc(%) |
|---|---|---|---|---|
| SANE | $71.84_{\pm 1.33}$ | $84.58_{\pm 0.53}$ | $87.55_{\pm 0.78}$ | $90.70_{\pm 0.89}$ |
| SANE* | $\mathbf{71.95}_{\pm 1.32}$ | $\mathbf{85.46}_{\pm 0.76}$ | $\mathbf{88.12}_{\pm 0.35}$ | $\mathbf{90.86}_{\pm 0.80}$ |

Table 4: Comparison of NAC with different initializations.

| | CiteSeer Acc(%) | Cora Acc(%) | PubMed Acc(%) | Computers Acc(%) |
|---|---|---|---|---|
| Kaiming Normal | $71.12_{\pm 2.45}$ | $86.85_{\pm 0.78}$ | $87.52_{\pm 0.72}$ | $91.05_{\pm 0.80}$ |
| Kaiming Uniform | $72.06_{\pm 2.24}$ | $86.99_{\pm 1.03}$ | $87.68_{\pm 0.97}$ | $88.52_{\pm 3.01}$ |
| Orthogonal | $\mathbf{74.62}_{\pm 0.38}$ | $\mathbf{87.41}_{\pm 0.92}$ | $\mathbf{88.04}_{\pm 1.06}$ | $\mathbf{91.64}_{\pm 0.14}$ |

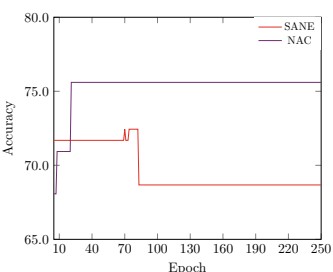

Figure 3: Convergence for SANE and NAC in terms of accuracy. NAC converges much faster than SNAE in only around 20 epochs.

experiment attempts to answer this by comparing NAC with and without training the output layer. Note that we fix all other layers with random weights. By default, NAC is the one without training any layers in the whole paper, including the output layer. On the contrary, NAC$^+$ is the one with training the output layer. Here, we set the number of epochs as 100. In Table. 2, we can find that the performance of NAC$^+$ is in the middle between SANE and NAC. On the one hand, training the linear layer may not be worthwhile because it is impossible to get the desired optimal weights. On the other hand, results with and without training the output layer are always better than SANE, suggesting the superiority of the no-update scheme in general. To further demonstrate this, we implement multiple experiments by training the last layer with different epochs, as shown in Appendix. A.4.3.

**Sparsity effect:** We also conduct experiments to test how sparsity affect the search quality of NAC. The hyperparameter $\rho$ is to control the sparsity of the architecture parameter $\boldsymbol{\alpha}$. We test the effect by varying its value in a considerably large range, i.e., $[0.001, 10]$. The results show that the performance varies little w.r.t. $\rho$. This indicates that NAC is insensitive to the sparsity hyperparameter.

**Random seeds:** By default, researchers report the average or the best result from different random seeds, this often lead to poor reproducibility. We now present the effect of random seeds on this topic. In particular, we implement the experiment on multiple random seeds and observe a stable performance of NAC.

The experimental results show that NAC is robust to both sparse hyperparameter selection and random seeds. Please refer to Appendix A.4 for more details.

## 4.5 ANALYSIS OF CONVERGENCE

A notable benefit of the NAC framework is its guaranteed convergence from the sparse coding perspective. To verify this, Fig. 3 offers a convergence comparison between NAC and SANE from the Pubmed dataset by showing the accuracy on the retrained network acquired at each training epoch. NAC shows substantially faster convergence than SANE, which take around 20 and 80 epochs, respectively. Due to the BLO strategy, SANE suffers strong oscillation when optimizing two variables, i.e., the architecture and neural weights, which consumes more epochs to converge.

## 5 CONCLUSION

We present the first linear complexity NAS algorithm for GNNs, namely NAC. Our NAC is solved by sparse coding and holds no-update scheme for model weights in GNNs because GNNs embed the built-in model linearity and orthogonality for the model weights. Extensive experiments demonstrate that our NAC can achieve higher accuracy (up to 18.8%) and much faster convergence (up to 200×) than SOTA NAS-GNNs baselines.

Several promising directions can be considered. We can further explore more deep neural networks that satisfy the mild condition of our NAC to extent its usability. We can investigate more on the efficiency of different subgradient methods for solving the sparse coding objective. We also intend to investigate how to jointly learn the search space and architectural representation to further enhance the expressive ability of searched architectures.

## REPRODUCIBILITY STATEMENT

The supplemental material includes the code for our experiments. An detailed description of the datasets used in the experiments is provided in Appendix A.1.

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

# A APPENDIX

## A.1 DATASET DETAILS

In the transductive learning task, node features and edges between nodes in the whole datasets were known beforehand. We learn from the already labeled training dataset and then predict the labels of the testing dataset. Three benchmark datasets were used in this setting: CiteSeerr (Giles et al., 1998), Cora (Kipf & Welling, 2016) and PubMed (Sen et al., 2008). All of the three benchmark datasets are citation networks. In citation networks, each node represents a work, and each edge shows the relationship between two papers in terms of citations. The datasets contain bag-of-words features for each node, and the goal is to categorize papers into various subjects based on the citation. In addition to the three benchmark datasets, we also employ another dataset Amazon Computers (McAuley et al., 2015). Amazon Computers is a subset of the Amazon co-purchase graph, where nodes represent commodities and edges connect them if they are frequently purchased together. Product reviews are encoded as bag-of-word feature vectors in node features, and class labels are assigned based on product category.

In the inductive task, several graphs are used as training data, while other completely unseen graphs are used as validation and test data. For the inductive setting, we use the PPI Hamilton et al. (2017) dataset as benchmark dataset. The task of PPI dataset is to classify the different protein functions. It consists of 24 graphs, with each representing a different human tissue. Each node has properties such as positional gene sets, motif gene sets, and immunological signatures, ant gene ontology sets as labels. Twenty graphs are chosen for training, two graphs for validation, and the remaining for testing. For the inductive task, we use Micro-F1 as the evaluation metric.

The used datasets are concluded in Table 5.

Table 5: The Statistics of Datasets

| | Transductive | | | | Inductive |
|---|---|---|---|---|---|
| | CiteSeer | Cora | PubMed | Computers | PPI |
| #nodes | 2708 | 3327 | 19717 | 13752 | 56944 |
| #edges | 5278 | 4552 | 44324 | 245861 | 818716 |
| #features | 1433 | 3703 | 500 | 767 | 121 |
| #classes | 7 | 6 | 3 | 10 | 50 |

## A.2 PROOF OF THEOREM 3.4 FOR THE DICTIONARY ORTHOGONALITY IN NAC

**Theorem A.1.** Let the neural weights of each operator in a deep neural network be an atom and stack them column-wisely, we can guarantee the existence of an orthogonal dictionary.

*Proof.* Given a dictionary $\boldsymbol{H} \in R^{n \times K}$, where $n$ is the number of nodes and $K$ is the number of opearators in each layer, we have its mutual coherence computed as follows,

$$\varphi = \max_{\boldsymbol{h}_i, \boldsymbol{h}_j \in \boldsymbol{H}, i \neq j} \left| \left\langle \frac{\boldsymbol{h}_i}{\|\boldsymbol{h}_i\|_2}, \frac{\boldsymbol{h}_j}{\|\boldsymbol{h}_j\|_2} \right\rangle \right|, \tag{8}$$

where $\varphi \in [0, 1]$, and $\langle \cdot \rangle$ denotes inner product. Here, each atom, $\boldsymbol{h}_i$, is the weights from an operator. The minimum of $\varphi$ is 0 and is attained when there is an orthogonal dictionary, while the maximum is 1 and it attained when there are at least two collinear atoms (columns) in a dictionary.

Let $E_i = \frac{\boldsymbol{h}_i}{\|\boldsymbol{h}_i\|_2}$ and $E_j = \frac{\boldsymbol{h}_j}{\|\boldsymbol{h}_j\|_2}$, by Central Limit Theorem (Fischer, 2011), we know that $\langle E_i, E_j \rangle / \sqrt{n}$ converges to a normal distribution, i.e.,

$$\langle E_i, E_j \rangle = \lim_{n \to \infty} \sqrt{n} Z, \tag{9}$$

where $Z$ is a standard normal distribution. Consider $\bar{E}$ as the mean value of all $\langle E_i, E_j \rangle$. With weak law of large numbers (a.k.a. Khinchin's law) (ter Haar, 1949), for any positive number $\varepsilon$, the probability that sample average $\bar{E}$ greater than $\varepsilon$ converges 0 is written as

$$\lim_{n \to \infty} \Pr\left(\left|\bar{E}\right| \geq \varepsilon\right) = 0 \tag{10}$$

This implies that the probability that the inner product of $E_i$ and $E_j$ is greater than $\varepsilon$ close to zero when $n \to \infty$. In other words, the probability that $E_i$ and $E_j$ are nearly orthogonal goes to 1 when their dimensionality is high. Therefore, the coherence of this dictionary reaches the minimum at a high dimensionality that holds for deep neural networks naturally. $\qquad \square$

### A.3 EXPERIMENTAL IMPLEMENTATION DETAILS

**Environment**. We implement experiments related to accuracy on Citeseer, Cora, and PubMed using PyTorch (Paszke et al., 2019) (version 1.10.2+cpu) on a CPU server that has a 48-core Intel Xeon Platinum 8260L CPU, and experiments on Amazon Computers using PyTorch (version 1.10.2+gpu) on a GPU server with four NVIDIA 3090 GPUs (24G). Speed-related experiments on Citeseer, Cora, and PubMed were measured using PyTorch (version 1.4+cpu) on a CPU server with a ten-core Intel Xeon Platinum 8255C CPU, 40G RAM, and 500G DRAM. Speed-related experiments on Amazon Computers were measured using PyTorch (version 1.10.2+gpu) on a GPU server with four NVIDIA 3090 GPUs (24G). Operators used in the experiments are from the built-in functions of PyG (version 2.0.2) (Fey & Lenssen, 2019).

**Searching configuration.** In our experiments we adopt 3-layer GNN as the backbone. Unless specified, our experiments follow the same settings for searching architectures as SANE (Zhao et al., 2021b) :

- *Architecture optimizer*. We use Adam for training the architecture parameters $\boldsymbol{\alpha}$. We set the learning rate as 0.0003 and the weight decay as 0.001. Also, the $\beta_1$ and $\beta_2$ are fixed as 0.5 and 0.999, respectively. All runs a constant schedule for training, such as 100 epochs.
- *Weight optimizer*. We use SGD to update models' parameters, i.e., $w$. The learning rate and SGD momentum are given as 0.025 and 0, respectively, where the learning rate has a cosine decay schedule for 100 epochs. We fix the weight decay value, i.e. set $\rho_1 = 0.0005$.
- *Batch size*. For transductive tasks, we adopt in-memory datasets, and the $batch\_size$ is fixed as the size of the dataset themselves.

**The configuration for retraining phase.** At the retraining state, we adopt Adam as the optimizer and set the scheduler with cosine decay to adjust the learning rate. The total number of epochs is fixed 400 for all methods for fairness. Please refer to the setting of SANE (Zhao et al., 2021b) and EGAN (Zhao et al., 2021a) for more details as we follow this in our experiment.

For CiteSeer dataset, we set the initial learning rate as 0.005937 and weight decay as 0.00002007. The configuration for models is as follows: $hidden\_size = 512$, $dropout = 0.5$, and using $ReLU$ as the activation function.

For Cora dataset, we set the initial learning rate as 0.0004150, and weight decay as 0.0001125. In model, we set $hidden\_size = 256$ , $dropout = 0.6$, and use $ReLU$ as the activation function.

For PubMed dataset, we set the initial learning rate as 0.002408 and weight decay as 0.00008850. As for the model, we have $hidden\_size = 64$ and $dropout = 0.5$, and use $ReLU$ as the activation function.

For Amazon dataset, we set the initial learning rate as 0.002111 and weight decay as 0.000331. As for the model, we have $hidden\_size = 64$ and $dropout = 0.5$, and use $elu$ as the activation function.

For PPI dataset, we set the initial learning rate as 0.00102 and weight decay as 0. As for the model, we have $hidden\_size = 512$ and $dropout = 0.5$, and use $Relu$ as the activation function.

**Solving $L^1$ regularization.** The $L^1$ regularization, also known as **Lasso Regression** (Least Absolute Shrinkage and Selection Operator), adds an absolute value of the magnitude of the coefficient as a penalty term to the loss function (Ranstam & Cook, 2018). Using the $L^1$ regularization, the coefficient of the less important feature is usually decreased to zero, sparsifying the parameters. It should be noted that since $||\boldsymbol{\alpha}||_1$ is not differentiable at $\boldsymbol{\alpha} = \mathbf{0}$, the standard gradient descent approach cannot be used.
Despite the fact that the loss function of the Lasso Regression cannot be differentiated, many approaches to problems of this kind, such as (Schmidt et al., 2009), have been proposed in the literature. These methods can be broadly divided into three groups: constrained optimization methods, unconstrained approximations, and sub-gradient methods.

Since subgradient methods are a natural generalization of gradient descent, this type of methods can be easily implemented in Pytorch's framework. Lasso Regression can be solved using a variety of subgradient techniques; details on their implementation can be found in (Fu, 1998) and (Shevade & Keerthi, 2003).

**Computational Complexity Estimation of NAC.** The computation of NAC has two major parts: the forward pass and the backward pass. Given the search space, the computation of the forward is then fixed and regarded as a constant. Therefore, the computational complexity mainly focuses on the backward pass in the NAC algorithms.

The main version of our work does not need to update weights, but only to update architectural parameter $\boldsymbol{\alpha}$ during the training process. Therefore, the algorithmic complexity is as $O(T * \|\boldsymbol{\alpha}\|)$, which is a *linear* function w.r.t $\boldsymbol{\alpha}$. The dimension of $\boldsymbol{\alpha}$ is often small, which makes the model easy to scale to large datasets and high search space. When updating weights of the linear layer, the complexity is estimated as $O(T * (\|\boldsymbol{\alpha}\| + \|\boldsymbol{W}_o\|))$. The dimension of $\boldsymbol{W}_o$ is a constant number, that equals the number of classes. Therefore, the complexity is almost the same as the main version of NAC, where the complexity is $O(T * \|\boldsymbol{\alpha}\| + \|\boldsymbol{W}_o\|)$.

When updating weights, similar to DARTS, the complexity is estimated as $O(T * (\|\boldsymbol{\alpha}\| + \|\boldsymbol{w}\|))$. The dimension of $\boldsymbol{w}$ is often much larger than $\boldsymbol{\alpha}$, therefore, the complexity is dominated by updating $\boldsymbol{w}$, where the complexity is $O(T * \|\boldsymbol{w}\|)$. Since the dimension of $\boldsymbol{\alpha}$ is much smaller than $\boldsymbol{w}$, the complexity of NAC is much less than this type of methods.

**Approximate Architecture Gradient.** Our proposed theorems imply an optimization problem with $\boldsymbol{\alpha}$ as the upper-level variable and $\boldsymbol{W}_o$ as the lower-level variable:

$$\begin{cases} \boldsymbol{\alpha}^* = \underset{\boldsymbol{\alpha}}{\operatorname{argmax}} \mathcal{M}\left(\boldsymbol{W}_o^*(\boldsymbol{\alpha}), \boldsymbol{\alpha}\right) \\ \boldsymbol{W}_o^*(\boldsymbol{\alpha}) = \underset{\boldsymbol{W}_o}{\operatorname{argmin}} \mathcal{L}(\boldsymbol{\alpha}, \boldsymbol{W}_o), \end{cases} \tag{11}$$

Following (Liu et al., 2019a), we can adopt a First-order Approximation to avoid the the expensive inner optimization, which allows us to give the implementation in the algorithm 1.

### A.4 ABLATION STUDIES

#### A.4.1 THE EFFECT OF SPARSITY

Our model uses the hyperparameter $\rho$ to control the sparsity of the architecture parameter $\boldsymbol{\alpha}$, where a large sparsity is to enforce more elements to be zero. We investigate the effect of this hyperparameter by varying its value in a considerably large range, such as $[0.001, 10]$. We first present the accuracy of different sparse setting in Fig. 4. We find that the results vary little in a considerably wide range, this indicates our models are insensitive to sparsity hyperparameter in general.

#### A.4.2 THE EFFECT OF RANDOM SEEDS

Random seeds often play an importance role in traditional NAS methods as it affects the initialization significantly. People often report the average or the best results under different random seeds, this may lead to poor reproducibility. To the best our knowledge, this is for the first we explicitly demonstrate the effect of random seeds in this subject. We run experiments on several random seeds and report the results of NAC on Pubmed dataset, as shown in Fig. 5. In particular, we implement multiple combinations of random seeds and sparsity to observe the variation on performance. Note that we round the values to integer to fit the table. In all these combinations, we have the average and variance as $87.32\%$ and $0.9\%$, respectively. The average performance is comparable to the best results from all competitive results, which indicates the stability of NAC.

#### A.4.3 THE EFFECT OF TRAINING ON THE FINAL LINEAR LAYER

Our proposed theorems prove that a GNN with randomly initialized weights can make the final output as good as a well-trained network when initializing networks with orthogonal weights and updating the total network using gradient descent. In practice, we find it difficult to determine at what training epoch the optimal weight parameters can be obtained through training linear layer. We noticed that most of the time, the untrained weights in the initial state can often already exceed the accuracy that can be obtained from the weights after multiple epochs of training the final linear layer, as shown in

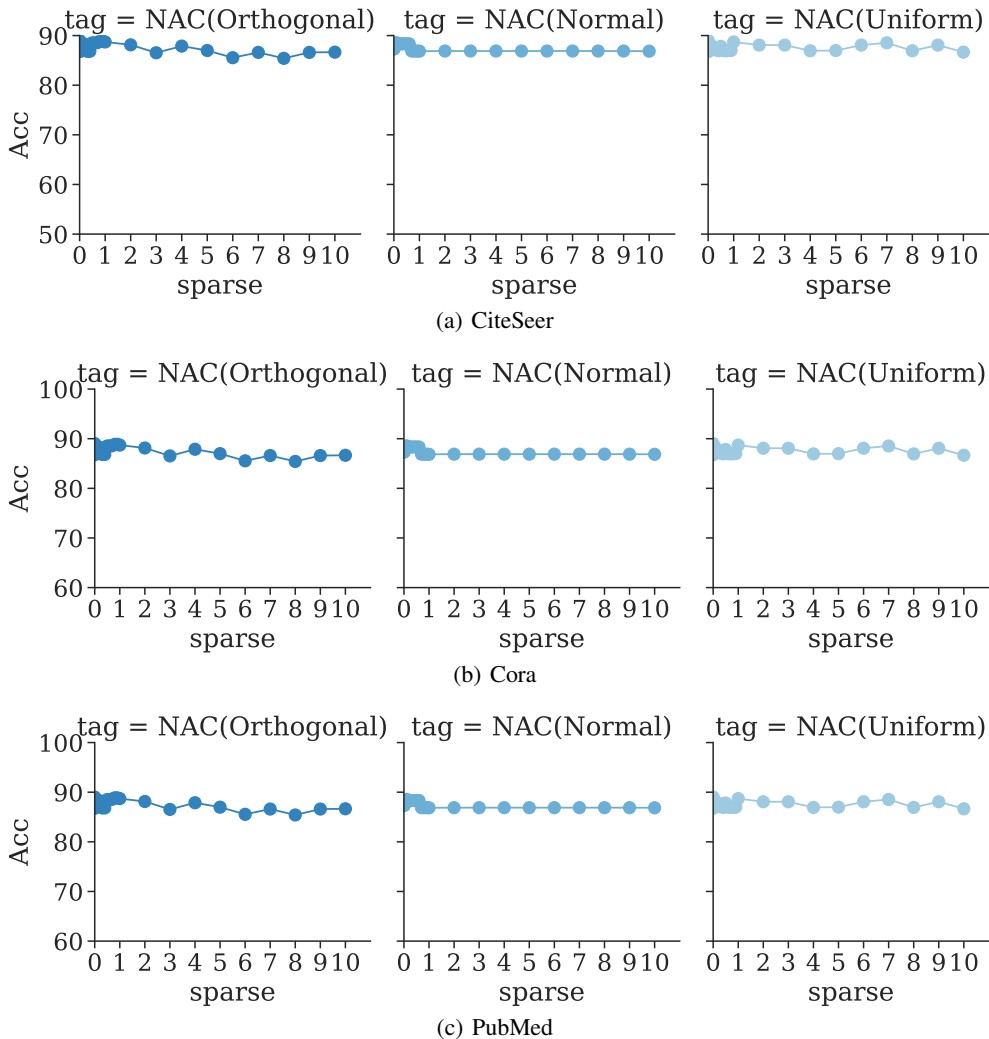

Figure 4: Sensitivity study of the sparsity. Results are with varying sparsity (x-axis) and different initialization NAC methods (i.e., normal, uniform and orthogonal). The variation is small, showing the robustness of our model w.r.t the sparsity.

Fig. 6. Therefore, we further omit the training of the final linear layer. It is important to note that this approximation is based on our proposed theorems in which most of the intermediate layers do not require training.

## A.5  RUNTIME OF EACH METHOD ON A SINGLE GPU SERVER

Apart from the running time on the CPU, we also measure the running time for all methods on a GPU platform, where we use PyTorch (version 1.10.2+gpu) on a GPU server with four NVIDIA 3090 GPUs (24G), as shown in Table 6. These results are consistent with the ones in Table 1, demonstrating our advantage in speed.

## A.6  PERFORMANCE ON INDUCTIVE TASKS

To validate the effectiveness of our approach to the Inductive task, we performed a set of experiments on the PPI dataset. The experimental results are concluded in Table 7. The experimental results show that our proposed method effectively outperforms the best baseline method by about 4% in terms of Micro-F1 score. Besides, our method achieves an $8\times$ speedup in terms of running time than SANE. Also, the non-updating scheme of the NAC approach exceeds the NAC-updating method effectively.

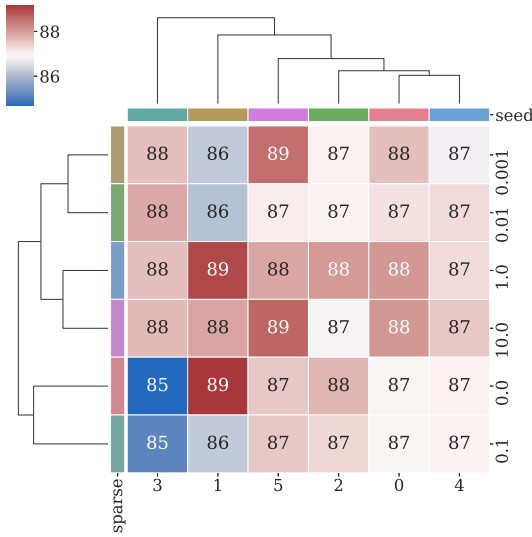

Figure 5: The effects of random seeds of NAC on the accuracy, where x-axis denotes the random seeds and y-axis denotes the sparsity. NAC performs stably with random seeds.

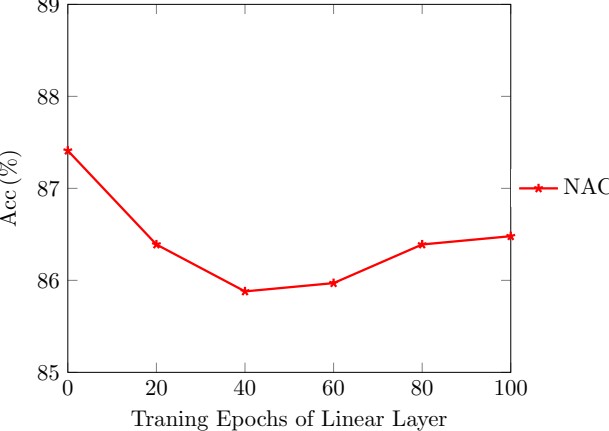

Figure 6: The effects of training final linear layer of NAC on the accuracy, where x-axis denotes the training epochs of the final linear layer and y-axis denotes the averaged accuracy of acquired architecture $\alpha$ using the corresponding weights.

Table 6: Timing results of the compared methods and our NAC method in one the same GPU server. NAC attains superior performance in efficiency (in seconds).

|          | CiteSeer | Cora    | PubMed  | Computers |
|----------|----------|---------|---------|-----------|
| RS       | 196.00   | 328.00  | 461.00  | 900.00    |
| BO       | 225.00   | 355.00  | 462.00  | 878.00    |
| GraphNAS | 6193.00  | 6207.00 | 6553.00 | 8969.00   |
| GraphNAS-WS | 947.00 | 1741.00 | 2325.00 | 4343.00  |
| SANE     | 35.00    | 41.00   | 43.00   | 43.00     |
| NAC      | **14.00**| **14.00**| **15.00**| **14.00**|
| NAC-updating | 42.00 | 31.00  | 36.00   | 42.00     |

Experiments on the PPI dataset further validate the effectiveness and superiority of our proposed method.

Table 7: Experimental results on the compared methods: our NAC attains superior performance on PPI dataset in both Micro-F1 score (%) and efficiency (in hours).

|          | PPI(Micro-F1(%)) | PPI(Time(h)) |
|----------|------------------|--------------|
| SANE     | $91.01_{\pm6.83}$ | 2.50 |
| NAC      | $\mathbf{95.16_{\pm0.03}}$ | **0.31** |
| NAC-updating | $94.47_{\pm7.09}$ | 4.68 |

## A.7 SEARCHED ARCHITECTURES FOR EACH DATASET OF OUR METHOD

We visualize the searched architectures (top-1) by NAC on different datasets in Fig.7.

- For Citeseer dataset, the searched result is GAT||GCN||Chebconv;
- For Cora dataset, the searched result is GIN||GIN||GCN;
- For Pubmed dataset, the searched result is GCN||GAT_Linear||Geniepath;
- For Amazon Computers dataset, the searched result is Geniepath||GCN||SAGE;
- For PPI dataset, the searched result is Chebconv||GAT_COS||Chebconv;

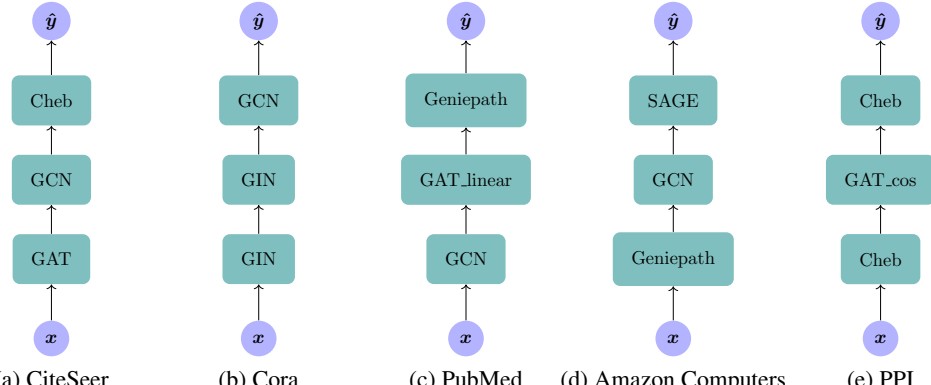

(a) CiteSeer     (b) Cora     (c) PubMed     (d) Amazon Computers     (e) PPI

Figure 7: The searched architectures of NAC on benchmark datasets.

