# OpenReview forum: "Do Not Train It: A Linear Neural Architecture Search of Graph Neural Networks"
_ICLR.cc/2023/Conference — Submitted to ICLR 2023_

### Official Review · Reviewer_WVaT · 2022-10-20

**Confidence:** 3
**Correctness:** 3
**Technical Novelty And Significance:** 2
**Empirical Novelty And Significance:** 2
**Recommendation:** 5

**Clarity, Quality, Novelty And Reproducibility:**

The paper is well constructed and clear.
The idea of using sparse coding for NAS in GNNs seems novel.

**Details Of Ethics Concerns:**

I haven't found any ethical issues.

**Strength And Weaknesses:**

*Strength*
The paper is interesting, and the proposed idea of using sparse coding for NAS in GNNs seems novel.
The proposed method seems to outperform previous art in terms of accuracy, taking less time.

*Weaknesses*
The paper presents the results only on transductive small datasets and does not provide a sota for comparison.
As a result, it is hard to understand the impact of the proposed method NAC.

**Summary Of The Paper:**

In this paper, the authors first provide theoretical proof that the untrained GNN models are nearly optimal,
second, propose a novel, based on sparse coding, NAS for GNNs: NAC - neural architecture coding.
Unlike other NAS methods, NAC, based on the assumption that untrained GNN models are nearly optimal, does not update GNN weights,
what leads to computational cost reduction and optimization stability improvement.

**Summary Of The Review:**

In general very interesting paper proposes new insights hoe to design NAS for GNNs in linear time. In addition, the paper is well constructed.
Even though there are a few weak points, which are summarized in the following questions:

1) Can you please elaborate on your statement that " GNN models behave almost linearly"?  I expect to see the theoretical or empirical justification for it.
The statement "Technical motivation for NAC stems from the observation that an untrained GNN performs well, and training it might not provide extra benefit"
fills like very strong and should be theoretically or empirically justified, preferable for both transductive and inductive datasets.

2) The proof result is "the output layer alone can ensure the optimality of the network even when all previous layers have no updates"
is strongly relied on skipping all nonlinear activation functions.
From my understanding, skipping all non-linear activations can dramatically reduce neural network feature extraction expressive power.
We can simply shrink all layers into a single layer, as a result, do not need depth.
Again, can authors provide empirical or theoretical justification for that?


3) From my understanding, the objection of NAS algorithms is to provide architectures that can achieve better accuracy than handcrafted architectures. I expect the authors also to add SOTA results of the evaluated datasets to justify the practical impact of the proposed method.

4) The experiments were only done on transductive datasets. It is known that inductive datasets are harder.
Can authors provide experimental results also for inductive datasets?

Pre-rebuttal score: I give 5 (BR), and looking forward to receiving the author's responses to my concerns.

---

> ### Author Response · Authors · 2022-11-14
> **Response to Reviewer WVaT**
>
> We thank Reviewer WVaT for reviewing our paper and providing helpful feedback on our work. Answers to some of your questions follow.
>
> > Can you please elaborate on your statement that "GNN models behave almost linearly"? I expect to see the theoretical or empirical justification for it. The statement "Technical motivation for NAC stems from the observation that an untrained GNN performs well, and training it might not provide extra benefit" fills like very strong and should be theoretically or empirically justified, preferable for both transductive and inductive datasets.
>
> **Answer**:
>
> - We have this statement based on existing literature, where [1] shows popular GNN models demonstrating similar linear structures, and other works like SGCN [4] also show linearity.
> - As we mentioned in the introduction, the first paper on modern GNNs, i.e., GCN [6], has already spotted the striking phenomenon of the untrained GNN model in the experiments. To explain this, for the first time, we provide a theoretical analysis in Section 3.1.
>
> > The proof result is "the output layer alone can ensure the optimality of the network even when all previous layers have no updates" is strongly relied on skipping all non-linear activation functions. From my understanding, skipping all non-linear activations can dramatically reduce neural network feature extraction expressive power. We can simply shrink all layers into a single layer, as a result, do not need depth. Again, can authors provide empirical or theoretical justification for that?
>
> **Answer**:
>
> - We only remove the activation function for theoretical analysis. With almost no exceptions, all existing research in GNNs ignores the activation function for analysis [1,2,3]. Otherwise, the analysis becomes almost impossible. For our proposed model, we still employ GNNs with activation functions as usual.
> - Indeed, under the required conditions, we can shrink all layers to a single layer since only the last layer matters, see Theorem 3 .1. We can hardly analyze the property of deep models without any constraints or assumptions. Therefore, the gap between theoretical analysis and implementation is inevitable, which is an open question for deep learning.
> - Based on our experiment, we witness an obvious advantage over the SOTA baseline, suggesting the effectiveness of what we claim in our theorems.
>
> > From my understanding, the objection of NAS algorithms is to provide architectures that can achieve better accuracy than handcrafted architectures. I expect the authors also to add SOTA results of the evaluated datasets to justify the practical impact of the proposed method.
>
> **Answer**:
> - Our work aims to show a simple but effective NAS-GNN paradigm. We proposed a theoretical guaranteed non-update policy to provide faster and more accurate outputs. Our method consistently outperforms SOTA NAS-GNN baselines under the same search space. The compared baselines, such as SANE, beat handcrafted architectures, so we skip the comparison with handcrafted GNNs.
> - Comparing with more SOTA GNNs is equivalent to expanding the search space, which is trivial and can be easily put into future work.
>
>
> > The experiments were only done on transductive datasets. It is known that inductive datasets are harder. Can authors provide experimental results also for inductive datasets?
>
> **Answer**:
>
> - We now add an experiment on the PPI dataset that has five more times nodes than others and get an obvious advantage when compared with the best-performing baseline, i.e., SANE[5].
>
> |  Methods   | PPI(Micro-F1(\%))  | PPI(Time(h)) |
> |  ----  | ----  | ----|
> | SANE | 91.01${_{\pm6.83}}$ | 2.50|
> | NAC | 95.16${_{\pm0.03}}$ | 0.31|
> | NAC-updating| 94.47${_{\pm7.09}}$| 4.68|
>
> **References**
>
> [1] Keyulu Xu etc. HOW POWERFUL ARE GRAPH NEURAL NETWORKS? ICLR 19.
>
> [2] Keyulu Xu etc. Optimization of Graph Neural Networks: Implicit Acceleration by Skip Connections and More Depth. ICML 21
>
> [3] Nian Liu etc. Revisiting Graph Contrastive Learning from the Perspective of Graph Spectrum. NIPS 22
>
> [4] Felix Wu etc. Simplifying graph convolutional networks. PMLR 2019
>
> [5] Huan Zhao, Quanming Yao, and Weiwei Tu. Search to aggregate neighborhood for graph neural network. In ICDE, 2021b.
>
> [6] Thomas N. Kipf, & Max Welling (2017). Semi-Supervised Classification with Graph Convolutional Networks. In International Conference on Learning Representations.

---

> ### Author Response · Authors · 2022-12-03
> **Response to Reviewer WVaT**
>
> Dear Reviewer WVaT,
>
> Thanks for your time and efforts in reviewing our paper. Regarding your questions, we append further experiments/explanations, most of which help to explain our article. We are eager to discuss the raised questions further. Please let us know if you still need clarification on any parts of our work.
>
> Best Regards,
>
> Paper 557 Authors

---

### Official Review · Reviewer_GxHc · 2022-10-20

**Confidence:** 4
**Correctness:** 3
**Technical Novelty And Significance:** 3
**Empirical Novelty And Significance:** 2
**Recommendation:** 5

**Clarity, Quality, Novelty And Reproducibility:**

The writing is clear. Evaluation is lacking a bit as only the simplest graph datasets are used.

**Strength And Weaknesses:**

Strength:
1. If the data assumption is correct, the proposed method does seem to achieve both better search results in much shorter time.
2. Some novelty in proposing the sparse coding reformulation of NAS for GNNs.

Weaknesses:
1. In theorem 3.2, the authors states the assumption: "data is linearly separable". I doubt this assumption holds true for all the graph datasets. Is there any proof that this is actually true?
2. Following the 1st point, GNN layers performing equally well without activation layers are usually observed on simpler GNN datasets using simple features such as Bag-of-words. I have some doubts that this will be the case when GNN datasets gets more complex and the interactions between nodes are more complex, e.g. in particle simulation or protein interaction datasets. In this case I am not sure if the proposed method can still work. Can authors perform experiment on more complex datasets?

Missing citation:
Zhao Y, Wang D, Gao X, et al. Probabilistic dual network architecture search on graphs[J]. arXiv preprint arXiv:2003.09676, 2020.

**Summary Of The Paper:**

The paper first proposed that on some graph datasets a model with untrained graph layers can perform equally well as trained graph layers, and then proposed a NAS method that exploits this trait to accelerate NAS training.

**Summary Of The Review:**

The proposed method works well but under a strong assumption and only on limited datasets. I can only recommend acceptance if authors can clarify my doubts above.

---

> ### Author Response · Authors · 2022-11-14
> **Response to Reviewer GxHc**
>
> We thank Reviewer GxHc for reviewing our paper and providing helpful feedback on our work. Answers to some of your questions follow.
>
> > In theorem 3.2, the authors state the assumption: "data is linearly separable ."I doubt this assumption holds true for all the graph datasets. Is there any proof that this is actually true?
>
> **Answer**:
>
> - The assumption of linearly separable is for the theoretical analysis. In real-world applications, the data is complicated and cannot guarantee its linearity. However, our experimental results show that our proposed NAC always attains competitive performance, which implies the robustness of our proposed NAC.
>
>
> > Following the 1st point, GNN layers performing equally well without activation layers are usually observed on simpler GNN datasets using simple features such as Bag-of-words.
>
> **Answer**:
>
> - We only remove the activation function for theoretical analysis. With almost no exceptions, all existing research in GNNs ignores the activation function for analysis [1,2,3]. Otherwise, the analysis becomes almost impossible. For our proposed model, we still employ GNNs with activation functions as usual.
>
> >  I have some doubts that this will be the case when GNN datasets get more complex and the interactions between nodes are more complex, e.g. in particle simulation or protein interaction datasets. In this case I am not sure if the proposed method can still work. Can authors perform experiment on more complex datasets?
>
> **Answer**:
>
> - In terms of evaluation, the data we experiment on follows the setup of the latest NAS-GNN papers, and our proposed method gains obvious advantages. We now add an experiment on the PPI dataset that has five more times nodes than others and get an obvious benefit when compared with the best-performing baseline, i.e., SANE [5].
>
> |  Methods   | PPI(Micro-F1(\%))  | PPI(Time(h)) |
> |  ----  | ----  | ----|
> | SANE | 91.01${_{\pm6.83}}$ | 2.50|
> | NAC | 95.16${_{\pm0.03}}$ | 0.31|
> | NAC-updating| 94.47${_{\pm7.09}}$| 4.68|
>
> > Missing citation: Zhao Y, Wang D, Gao X, et al. Probabilistic dual network architecture search on graphs[J]. arXiv preprint arXiv:2003.09676, 2020.
>
> **Answer**:
>
> - We appreciate the reviewer's suggestion and will add the paper to the related work.
> - We have added the related work as in Sec. 1 paragraph one, which is highlighted in blue color.
>
> **References**
>
> [1] Keyulu Xu etc. HOW POWERFUL ARE GRAPH NEURAL NETWORKS? ICLR 19.
>
> [2] Keyulu Xu etc. Optimization of Graph Neural Networks: Implicit Acceleration by Skip Connections and More Depth. ICML 21
>
> [3] Nian Liu etc. Revisiting Graph Contrastive Learning from the Perspective of Graph Spectrum. NIPS 22
>
> [4] Felix Wu etc. Simplifying graph convolutional networks. PMLR 2019
>
> [5] Huan Zhao, Quanming Yao, and Weiwei Tu. Search to aggregate neighborhood for graph neural network. In ICDE, 2021b.

---

> ### Author Response · Authors · 2022-12-03
> **Response to Reviewer GxHc**
>
> Dear Reviewer GxHc,
>
> Thanks for your time and efforts in reviewing our paper. Regarding your questions, we append further experiments/explanations, most of which help to explain our article. We are eager to discuss the raised questions further. Please let us know if you still need clarification on any parts of our work.
>
> Best Regards,
>
> Paper 557 Authors

---

### Official Review · Reviewer_oy8p · 2022-10-24

**Confidence:** 3
**Correctness:** 3
**Technical Novelty And Significance:** 2
**Empirical Novelty And Significance:** 3
**Recommendation:** 5

**Clarity, Quality, Novelty And Reproducibility:**

It is interesting and novel to utilize the no-update GNNs in NAS-GNN, while some aspects are unclear.

**Strength And Weaknesses:**

### Strength:

This paper shows the feasibility of no-update GNNs in graph representation learning, and then applied them into NAS-GNN, which can transfer the bi-level optimization problem to single-level. It is interesting and useful for NAS-GNN.

### Weakness:

1. The optimization gap still exists due to the shared output layer.
In Theorem 3.2, one untrained GNN (which is denoted as A^LXO and omits the output layer) can make the final output as good as the well-trained GNNs based on the optimized weights $\hat{W}^*_o$. It seems that this weight still needs to be trained with the first term of the optimization objective in Eq.(7). Then, the optimization gap cannot be addressed since different architectures also share these untrained weights (W_l) in each layer and the output layer(W_o). When we obtained the optimized weight $\hat{W}^*_o$ with Eq.(7), this weight is sub-optimal for each candidate architecture in the search space. Therefore, the optimization gap still exists.

2. The constraints on the dictionary D in sparse encoding. The second term in Eq.(3) is ignored in Eq.(7).

3. Some experiments are ignored.
a) This paper only considered the NAS baselines in GNNs. Compared with SANE(Zhao et al. 2021b), the baseline performance is pretty lower.

b) Since this paper remove all the non-linear operations in GNNs. The untrained SGC baseline should be considered. It seems that it is one competitive baseline with Theorem 3.1-3.3.

c) The evaluations on “complex” datasets. As mentioned in Page 8, “the optimal condition can get compromised due to the high complexity of the data.” It seems that the proposed method has difficulty when applied to a complex dataset. However, how to measure the “complexity” of the dataset? Besides, the proposed method aims to learn the sparse encoding for each node, which is equivalent to designing the node-wise GNNs. It would be better to evaluate the proposed procedures, e.g., large-scale datasets ogbn-arxiv.


**Summary Of The Paper:**

This paper proposed one method NAC to automatically design GNNs. With randomly initialized model weights, NAC designs the GNNs by learning the sparse encodings on top of the search space. The empirical and theoretical results demonstrate its effectiveness.

**Summary Of The Review:**

This paper proposed one method NAC to automatically design GNNs based on the no-update GNNs. The theoretically analysis and some experiments can be further improved.

---

> ### Author Response · Authors · 2022-11-14
> **Response to Reviewer oy8p (1/2)**
>
> We thank Reviewer oy8p for reviewing our paper and providing helpful feedback on our work. Thank you for pointing out these issues, and we would like to take this chance to clarify them.
>
> > The optimization gap still exists due to the shared output layer. In Theorem 3.2, one untrained GNN (which is denoted as $A^LXO$ and omits the output layer) can make the final output as good as the well-trained GNNs based on the optimized weights $\hat{W}_{o}^{∗}$.
>
> > It seems that this weight still needs to be trained with the first term of the optimization objective in Eq.(7). Then, the optimization gap cannot be addressed since different architectures also share these untrained weights ($W_l$) in each layer, and the output layer($W_o$) when we obtained the optimized weight $\hat{W}_{o}^{∗}$ with Eq.(7), this weight is sub-optimal for each candidate architecture in the search space. Therefore, the optimization gap still exists.
>
> **Answer**:
>
> - Our method and all other weight-sharing (WS) methods share this optimization gap, but this does affect the result as the goal is not to get optimal weights in the architecture searching stage.
> - Recall that NAS-GNN methods consist of two stages: training supernet is to obtain the optimal architecture, and retraining is to get the optimal weight for the obtained architecture.
> In the first stage, we do not require the candidate architecture to have its optimal weights. Obtaining optimal weights directly for a candidate architecture is one way to select architectures. However, this is too time-consuming and only used in early NAS-GNN methods. Recent work has focused on the weight-sharing (WS) paradigm, including ours, to avoid this situation.
> - Unlike other WS methods, we provide a theoretical guarantee to ensure the selection needs no training as opposed to the early-stopping-based empirical strategy in prior WS methods. After obtaining the optimal architecture, in the second stage, we will train its weight to the optimum, called retraining.
>
> > The constraints on the dictionary D in sparse encoding. The second term in Eq.(3) is ignored in Eq.(7).
>
> **Answer**:
>
> - The second term is the hard threshold of sparsity, and one naive implementation of sparsity control is hard thresholding. In our work, we chose to control the hyperparameter in training and take the top-ranked elements to get the sparse output. Besides, GNN layers already include a normalization process, and we do not need to perform explicit normalization.
>
> > Some experiments are ignored. a) This paper only considered the NAS baselines in GNNs. Compared with SANE[5], the baseline performance is pretty lower.
>
> **Answer**:
>
> - For question a). The scope of this paper is to develop a new paradigm for NAS-GNNs. Therefore we compare against related baselines from this domain.
> - For problem (b), the performance of SANE [5] is inferior to that in the original paper due to the following reasons.
>     - First, we do not include the JK operator in the search space since it contradicts our assumption, i.e., JKNet is not in a matrix multiplication format.
>     - Second, we do not apply its hyperparameter tuning process because this is unfair to other baselines.
>
> > Since this paper remove all the non-linear operations in GNNs. The untrained SGC baseline should be considered. It seems that it is one competitive baseline with Theorem 3.1-3.3.
>
> **Answer**:
>
> - We do not compare GNNs with SGC because our model employs GNNs with the activation function.
> - We only remove the activation function for theoretical analysis. This follows the setup of the analysis in [1,2.3]. In the experiments, we still employ GNNs with activation functions as usual.
>
> > The evaluations on "complex" datasets. As mentioned on Page 8, "the optimal condition can get compromised due to the high complexity of the data." It seems that the proposed method has difficulty when applied to a complex dataset. However, how to measure the "complexity" of the dataset?
>
> **Answer**:
>
> - As stated in Section 3.1, we mention the complexity in data is to be honest with the condition of linearly separable, which is to make the theorem rigorous. As we all know, real-world data is complicated; therefore, we cannot guarantee that any given data is linearly separable.
> - Our experiments, however, always show decent performance, indicating the robustness of our model even under the potential of compromising this condition.
>
>
> **References**
>
> [1] Keyulu Xu etc. HOW POWERFUL ARE GRAPH NEURAL NETWORKS? ICLR 19.
>
> [2] Keyulu Xu etc. Optimization of Graph Neural Networks: Implicit Acceleration by Skip Connections and More Depth. ICML 21
>
> [3] Nian Liu etc. Revisiting Graph Contrastive Learning from the Perspective of Graph Spectrum. NIPS 22
>
> [4] Felix Wu etc. Simplifying graph convolutional networks. PMLR 2019
>
> [5] Huan Zhao, Quanming Yao, and Weiwei Tu. Search to aggregate neighborhood for graph neural network. In ICDE, 2021b.

---

> > ### Author Response · Authors · 2022-11-14
> > **Response to Reviewer oy8p (2/2)**
> >
> > >  Besides, the proposed method aims to learn the sparse encoding for each node, which is equivalent to designing the node-wise GNNs. It would be better to evaluate the proposed procedures, e.g., large-scale datasets ogbn-arxiv.
> >
> > **Answer**:
> > - Our goal is not to utilize sparse coding to obtain node-wise GNNs but to obtain an optimal architecture that combines GNN operators optimally. We convert the architecture search problem into a sparse coding problem. After obtaining the best architecture, the rest are the same as other hand-crafted GNNs.
> > - As to the selection of datasets, we follow the setup of previous NAS-GNN methods. We now add an experiment on the PPI dataset that has five more times nodes than others and get an obvious advantage when compared with the best-performing baseline, i.e., SANE [5].
> >
> > |  Methods   | PPI(Micro-F1(\%))  | PPI(Time(h)) |
> > |  ----  | ----  | ----|
> > | SANE | 91.01${_{\pm6.83}}$ | 2.50|
> > | NAC | 95.16${_{\pm0.03}}$ | 0.31|
> > | NAC-updating| 94.47${_{\pm7.09}}$| 4.68|
> >
> > **References**
> >
> > [1] Keyulu Xu etc. HOW POWERFUL ARE GRAPH NEURAL NETWORKS? ICLR 19.
> >
> > [2] Keyulu Xu etc. Optimization of Graph Neural Networks: Implicit Acceleration by Skip Connections and More Depth. ICML 21
> >
> > [3] Nian Liu etc. Revisiting Graph Contrastive Learning from the Perspective of Graph Spectrum. NIPS 22
> >
> > [4] Felix Wu etc. Simplifying graph convolutional networks. PMLR 2019
> >
> > [5] Huan Zhao, Quanming Yao, and Weiwei Tu. Search to aggregate neighborhood for graph neural network. In ICDE, 2021b.

---

### Official Review · Reviewer_9TQX · 2022-10-25

**Confidence:** 4
**Correctness:** 3
**Technical Novelty And Significance:** 3
**Empirical Novelty And Significance:** 3
**Recommendation:** 6

**Clarity, Quality, Novelty And Reproducibility:**

The motivation of this work is quite novel. And the presentation of this paper is well organized. Besides, the authors add their code to supplementary material.

**Strength And Weaknesses:**

Strength:
1. The proposed NAC is 200x faster than existing GNN-NAS methods.
2. Besides, the NAC can find better GNN architectures than existing GNN-NAS methods.
3. This work has a good theoretical contribution.

Weaknesses:
1. The assumption is strong and the theory holds for GNNs without activation functions.
2. There is a lack of experiments on large-scale graph data.
3. The contrast of SOTA GNN is missing in the experiment.
4. The GNN architecture searched by the proposed method is not shown.

**Summary Of The Paper:**

This work firstly proved that GNNs (without activation function) fixed with random weights can make final outputs as good as well-trained GNNs under mild conditions. Based on this foundation, the authors proposed a novel graph neural architecture search method neural architecture coding (NAC). Specifically, NAC holds a no-update scheme on the parameters of GNNs and concentrates on the training of architecture parameters. And the NAC is 200x faster than existed GNN-NAS methods and can find better GNN architecture.

**Summary Of The Review:**

    I recommend acceptance. This work proposed a linear complexity GNN-NAS method NAC. And the proposed NAC faster than existed GNN-NAS methods and can find GNNs with better performance than existing GNN-NAS methods. What’s more, the work has a good theoretical foundation. Here are my major concerns:

    1. The conditions of Theorem 3.1 and Theorem 3.3 are very strong. They only apply to some GNNs without activation functions.
    2. There are errors in servel statement of this paper.  “Graph neural networks behave almost linearly, so they can be simplified as linear networks while maintaining superior performance (Wu et al.,2019b).” They only claim GCN can be simplified as linear networks.
    3. Considering that the search space is different from the previous methods, it is suggested to add ablation experiments to verify the impact of the search space.
    4. It is suggested to add training-free NAS methods to related works, such as [1][2].
    5.  It is suggested to add SOTA GNN methods to related works.

[1] Neural architecture search on imagenet in four gpu hours: A theoretically inspired perspective
[2] NASI: Label-and Data-agnostic Neural Architecture Search at Initialization

---

> ### Author Response · Authors · 2022-11-14
> **Response to Reviewer 9TQX**
>
> We thank Reviewer 9TQX for reviewing our paper and providing helpful feedback on our work. Answers to some of your questions follow.
>
> > The conditions of Theorem 3.1 and Theorem 3.3 are very strong. They only apply to some GNNs without activation functions.
>
> **Answer**:
>
> - Though removing the activation function to simplify GNNs analysis is a strong assumption in Theorem 3.1 and Theorem 3.3, we argue that this is a common setup for the analysis; see, e.g., in [1,2,3]. Hence, we follow the standard setup in our theoretical analysis.
>
> > There are errors in the servel statement of this paper. "Graph neural networks behave almost linearly, so they can be simplified as linear networks while maintaining superior performance (Wu et al.,2019b)." They only claim GCN can be simplified as linear networks.
>
> **Answer**:
>
> - Our statement "... linear ..." comes from a well-known theoretical result in [1] that many GNNs share the similar linear structure, and partially from [5] that uses linearity to analyze non-linear networks. Hence, we make a mild assumption to analyze the properties of NAC on GNNs.
>
> > It is suggested to add SOTA GNN methods to related works.
>
> **Answer**:
>
> - In our experiment, we have included popular GNNs by following the setup in the latest NAS-GNN work: SANE[4].
> - Our experimental results have demonstrated the efficiency and effectiveness of our model when compared with SOTA NAS-GNN methods. Compared with more SOTA GNNs is equivalent to expanding the search space, which is trivial and can be easily put into future work.
> - Besides, we add an experiment on the PPI dataset that has five more times nodes than others and get an obvious advantage when compared with the best-performing baseline, i.e., SANE [4].
>
> |  Methods   | PPI(Micro-F1(\%))  | PPI(Time(h)) |
> |  ----  | ----  | ----|
> | SANE | 91.01${_{\pm6.83}}$ | 2.50|
> | NAC | 95.16${_{\pm0.03}}$ | 0.31|
> | NAC-updating| 94.47${_{\pm7.09}}$| 4.68|
>
> > It is suggested to add training-free NAS methods to related works.
>
> **Answer**:
>
> - We thank the reviewer's suggestion and will add more papers to the related work.
> - We have added the related work as in Sec. 2 paragraph two, which is highlighted in blue color.
>
> > The GNN architecture searched by the proposed method is not shown.
>
> **Answer**:
>
> Please note that we have put the search results in Appendix A.6. Here, we list some examples as follows:
>
> 1. For the Citeseer dataset, the searched result is `GAT||GCN||Chebconv`;
> 2. For the Cora dataset, the searched result is `GIN||GIN||GCN`;
> 3. For the Pubmed dataset, the searched result is `GCN||GAT_Linear||Geniepath`;
> 4. For Amazon Computers dataset, the searched result is `Geniepath||GCN||SAGE`;
> 5. For the PPI dataset, the searched result is `Chebconv||GAT_COS||Chebconv`;
>
> **References**
>
> [1] Keyulu Xu etc. HOW POWERFUL ARE GRAPH NEURAL NETWORKS? ICLR 19.
>
> [2] Keyulu Xu etc. Optimization of Graph Neural Networks: Implicit Acceleration by Skip Connections and More Depth. ICML 21
>
> [3] Nian Liu etc. Revisiting Graph Contrastive Learning from the Perspective of Graph Spectrum. NIPS 22
>
> [4] Huan Zhao, Quanming Yao, and Weiwei Tu. Search to aggregate neighborhood for graph neural network. In ICDE, 2021b.
>
> [5] Xu K, Zhang M, Jegelka S, et al. Optimization of graph neural networks: Implicit acceleration by skip connections and more depth[C]//International Conference on Machine Learning. PMLR, 2021: 11592-11602.

---

> ### Author Response · Authors · 2022-12-03
> **Response to Reviewer 9TQX**
>
> Dear Reviewer 9TQX,
>
> Thanks for your time and efforts in reviewing our paper. Regarding your questions, we append further experiments/explanations, most of which help to explain our article. We are eager to discuss the raised questions further. Please let us know if you still need clarification on any parts of our work.
>
> Best Regards,
>
> Paper 557 Authors

---

### Official Review · Reviewer_P1Y6 · 2022-11-04

**Confidence:** 4
**Clarity, Quality, Novelty And Reproducibility:** 1. I think the clarity of the center …
**Correctness:** 3
**Technical Novelty And Significance:** 3
**Empirical Novelty And Significance:** 3
**Recommendation:** 5

**Strength And Weaknesses:**

#### Strengths
1. I like the idea of linearizing the GNNs for the NAS problem and formulating the search of architecture parameters as a sparse coding problem. To my best knowledge, this should be novel in the line of research of NAS for GNNs.
2. The empirical results look promising compared to the previous methods in terms of performance and efficiency.

#### Weaknesses
1. One major concern of mine is regarding the clarity of the theory & algorithm parts. I think the major contribution of this work is the NAC algorithm which is described in section 3.2. However, I think the clarity (and length) of section 3.2 could be much improved. In terms of length, I think section 3.1 takes too much space, and I suggest moving the detailed proofs to the appendices. In terms of clarity, I suggest the authors define and describe the concepts/notations with more details. For example, how do we encode/what is the dimensionality of the GNN operator $\alpha$; is $o^l$ a set function (aggregator) or a normal vector function (as the last line of Eq.(7))? The Eq. (7) is the central contribution and should be explained and analyzed with much greater details. Moreover, the notations and logic in sections 3.1 and 3.2 are somehow disconnected, and it is also hard to understand how is the theorems in section 3.1 useful to the NAC algorithm.
2. The proposed NAC method outputs the optimal architecture by an argmax in each layer in Algorithm 1, so I think this method should be better understood as a differentiable NAS or Auto-GNN method. Does the author ensure the search space is exactly the same among all baselines, including random search (RS), Bayesian optimization (BO), and GraphNAS in Table 1? Also, is it possible to also consider Auto-GNN (and maybe some newer SOTA methods in that line of research) as a fair baseline?
3. The evaluation is mostly done on very small graphs, and this is not sufficient to justify the (consistent) advantage of the proposed method on large-scale real-world graphs. Moreover, this paper also claims the efficiency advantages. Thus it would be necessary to evaluate on at least a few larger graphs.


#### Minor Issues
1. The format of the uploaded PDF is not directly generated from LaTeX, i.e., it seems that each page is an image thus, I cannot search/select text, or use the hyperlinks.

**Summary Of The Paper:**

This paper considers the neural architecture search (NAS) problem specifically for graph neural networks (GNNs). By omitting the effect of non-linearity of GNNs (in terms of the NAS ranking of architectures), i.e., we can linearize the GNNs, and the NAS for GNN problem can be greatly simplified and formulated as a sparse encoding problem, which this paper called neural architecture coding (NAC). This paper first proves the linearized GNN can still achieve optimality under mild conditions (Theorem 3.1-3.3), and then proposes to search/optimize the architecture parameters via the sparse coding method. Experiments show that the proposed NAC method is much faster and accurate than some previous NAS for GNNs baselines.

**Summary Of The Review:**

Overall for the current manuscript, I recommend weak rejection. I do like the idea of NAC and agree with its novelty. But I think some rewrites are needed to improve the clarity, especially about the description of the NAC algorithm and related analysis, which could be extended with more details. Moreover, the experiments may also need a bit more recent baselines from the Auto-GNN line of research and, more importantly, evaluation on at least a few larger graphs to support the efficiency claims.

---

> ### Author Response · Authors · 2022-11-14
> **Response to Reviewer P1Y6**
>
> We thank Reviewer P1Y6 for reviewing our paper and providing helpful feedback on our work. In this paper, we show that it is possible to approach the optimal network weights by applying random initialization (section 3.1) and propose our method (section 3.2) to fit our theorems, making it the first linear NAS-GNNs method. We are sorry for the confusion of this organization and now add more related details in the paper. Here are a few examples regarding your questions:
>
> > In terms of clarity, I suggest the authors define and describe the concepts/notations with more detail. For example, how do we encode/what is the dimensionality of the GNN operator $\alpha$; is $o^l$ a set function (aggregator) or a normal vector function (as the last line of Eq.(7))? The Eq. (7) is the central contribution and should be explained and analyzed with much greater details.
>
> **Answer**:
>
> - The hidden size of each operator is the same as the dimensionality of the GNN operator, and operators in each layer share the same hidden dimension.
> - $o^l$ is an operator, and its output is a vector with the same size as the hidden dimension.
> - $\alpha$ is the sparse coefficient vector for the operators indicating the importance of each operator.
>
> > Moreover, the notations and logic in sections 3.1 and 3.2 are somehow disconnected, and it is also hard to understand how is the theorems in section 3.1 useful to the NAC algorithm.
>
> **Answer**:
>
> - As we stated in the paper, it is possible to approach the optimal network weights by applying random initialization (section 3.1) and propose our method (section 3.2) to fit our theorems, making it the first linear NAS-GNNs method.
> - We have highlighted the changed parts in blue color in the revision.
>
> > The reproducibility depends on whether the authors will release the code (conditioned on the acceptance) and cannot be judged now.
>
> **Answer**:
>
> - We have included the code in the appendix in the submission, including training and evaluation. We now provide a link as a backup: [NAC.codes](https://drive.google.com/file/d/1wWR8vhF0JB0EbFahLWLct4yyNX5oAc8m/view?usp=sharing)
>
> > Does the author ensure the search space is exactly the same among all baselines, including random search (RS), Bayesian optimization (BO), and GraphNAS in Table 1? Also, is it possible to also consider Auto-GNN (and maybe some newer SOTA methods in that line of research) as a fair baseline?
>
> **Answer**:
>
> - Our experiments ensure that all methods have the same search space.
> - According to the learning strategy, GraphNAS[1] and Auto-GNN[3] belong to the same type and show nearly identical performance. Hence, we select one of them, i.e., GraphNAS[1], as a strong baseline. Besides, the SANE[2] baseline is one of the latest SOTA NAS-GNNs. That is also why we choose to compare against it.
>
> > The evaluation is mostly done on very small graphs, and this is not sufficient to justify the (consistent) advantage of the proposed method on large-scale real-world graphs. Moreover, this paper also claims the efficiency advantages. Thus it would be necessary to evaluate on at least a few larger graphs.
>
> **Answer**:
>
> - The data we experiment on follows the setup of previous NAS-GNNs, such as AutoGNN[3], GraphNAS[2], and SANE[1].
> - Our complexity is linear because no neural weight update is needed, resulting in the first linear model of NAS-GNNs and thus has an absolute advantage in theory rather than empirical study.
> - Besides, we add an experiment on the PPI dataset that has five more times nodes than others and get an obvious advantage when compared with the best-performing baseline, i.e., SANE[1].
>
> |  Methods   | PPI(Micro-F1(\%))  | PPI(Time(h)) |
> |  ----  | ----  | ----|
> | SANE | 91.01${_{\pm6.83}}$ | 2.50|
> | NAC | 95.16${_{\pm0.03}}$ | 0.31|
> | NAC-updating| 94.47${_{\pm7.09}}$| 4.68|
>
>
> > The format of the uploaded PDF is not directly generated from LaTeX, i.e., it seems that each page is an image thus, I cannot search/select text, or use the hyperlinks.
>
> **Answer**:
>
> - We appreciate the pointing and are sorry for the inconvenience caused. The latest updated version has fixed this.
>
>
> **References**
>
> [1] Yang Gao, Hong Yang, Peng Zhang, Chuan Zhou, and Yue Hu. Graph neural architecture search. In IJCAI, 2020.
>
> [2] Huan Zhao, Quanming Yao, and Weiwei Tu. Search to aggregate neighborhood for graph neural network. In ICDE, 2021b.
>
> [3] Kaixiong Zhou, Qingquan Song, Xiao Huang, and Xia Hu. Auto-gnn: Neural architecture search of graph neural networks. CoRR, abs/1909.03184, 2019.

---

> ### Author Response · Authors · 2022-12-03
> **Response to Reviewer P1Y6**
>
> Dear Reviewer P1Y6,
>
> Thanks for your time and efforts in reviewing our paper. Regarding your questions, we append further experiments/explanations, most of which help to explain our article. We are eager to discuss the raised questions further. Please let us know if you still need clarification on any parts of our work.
>
> Best Regards,
>
> Paper 557 Authors

---

### Author Response · Authors · 2022-11-14
**Response to all reviewers**

We appreciate all reviewers for your responsible and insightful feedback.  We would like to provide clarity for a few common questions.

1. We remove the activation function in Section 3.1 because this is a common setup for the theoretical analysis of GNNs. Analyzing the behavior of deep learning models is an open question due to its nonlinearity, such as [1][2][3]. With almost no exceptions, all existing research in GNNs ignores the activation function for analysis.  Note that we use standard GNNs with activation functions for modeling and experiments.
2. This paper revives the power of the non-updating mechanism in GNNs, which was discovered by the first modern GCN paper [4] but has received no attention since then. To the best of our knowledge, our work is the first to unveil this officially and provide theoretical proof.
3.  Our work aims to show a simple but effective NAS-GNN paradigm, and our method consistently outperforms SOTA NAS-GNN baselines under the same search space. Hence, comparison with more SOTA GNNs is equivalent to expanding the search space, which is trivial and can be performed in future work easily.


**References**
[1] Keyulu Xu etc. HOW POWERFUL ARE GRAPH NEURAL NETWORKS? ICLR 19.
[2] Keyulu Xu etc. Optimization of Graph Neural Networks: Implicit Acceleration by Skip Connections and More Depth. ICML 21
[3] Nian Liu etc. Revisiting Graph Contrastive Learning from the Perspective of Graph Spectrum. NIPS 22
[4] Thomas N. Kipf, & Max Welling (2017). Semi-Supervised Classification with Graph Convolutional Networks. In International Conference on Learning Representations.

---

### Decision · Program_Chairs · 2023-01-20

**Decision:**

Reject

**Justification For Why Not Higher Score:**

Although the idea of formulating the NAS problem for linearized GNN as a sparse encoding problem is interesting, the clarity of this paper needs much improvement. The authors explained some notations to the reviewers and added them to the paper. But reviewers agree that these changes are not enough to significantly improve the readability/accessibility of the paper. Some of the parts of the paper require much more discussion and background. One example is Theorem 3.4, Eq. (7). Another example is the "linearization of the GNN" which is explained as "We remove the activation function in Section 3.1 because this is a common setup for the theoretical analysis of GNNs". Some reviewers still have doubts about this (e.g., 2nd point from reviewer 9TQX and WVaT). Reviewers expect more discussion & justification about this instead of further shortening/removing the descriptions.

As a method for efficient NAS on GNNs, this paper did not consider any large-enough or medium-large size graphs (three reviewers share this concern). The largest graph used is the PPI dataset which has around 40K nodes, still a small-scale dataset (much smaller than any OGB node classification graphs).

There is a concern on the "linearization of the GNN" proof. While the authors relied on SGCN for the justification, the theoretical analysis can be more solid.

**Justification For Why Not Lower Score:**

N/A

**Metareview: Summary, Strengths And Weaknesses:**

This paper proposes linearized GNNs for the neural architecture search (NAS) problem on graph neural networks (GNNs). The authors simplify the NAS for GNN problem and formulate it as a sparse encoding problem, called neural architecture coding (NAC) in this paper. The authors prove the linearized GNN's optimality under mild conditions. The paper also proposes to search/optimize the architecture parameters via the sparse coding method.

- Although the idea of formulating the NAS problem for linearized GNN as a sparse encoding problem is interesting, the clarity of this paper needs much improvement. The authors explained some notations to the reviewers and added them to the paper. But reviewers agree that these changes are not enough to significantly improve the readability/accessibility of the paper. Some of the parts of the paper require much more discussion and background. One example is Theorem 3.4, Eq. (7). Another example is the "linearization of the GNN" which is explained as "We remove the activation function in Section 3.1 because this is a common setup for the theoretical analysis of GNNs". Some reviewers still have doubts about this (e.g., 2nd point from reviewer 9TQX and WVaT). Reviewers expect more discussion & justification about this instead of further shortening/removing the descriptions.

- As a method for efficient NAS on GNNs, this paper did not consider any large-enough or medium-large size graphs (three reviewers share this concern). The largest graph used is the PPI dataset which has around 40K nodes, still a small-scale dataset (much smaller than any OGB node classification graphs).

- There is a concern on the "linearization of the GNN" proof. While the authors relied on SGCN for the justification, the theoretical analysis can be more solid.